# Thermal Cycling-Hyperthermia Attenuates Rotenone-Induced Cell Injury in SH-SY5Y Cells Through Heat-Activated Mechanisms

**DOI:** 10.3390/ijms26146671

**Published:** 2025-07-11

**Authors:** Yu-Yi Kuo, Guan-Bo Lin, You-Ming Chen, Hsu-Hsiang Liu, Fang-Tzu Hsu, Yi Kung, Chih-Yu Chao

**Affiliations:** 1Department of Physics, Laboratory for Medical Physics & Biomedical Engineering, National Taiwan University, Taipei 106319, Taiwan; yykuo@phys.ntu.edu.tw (Y.-Y.K.); gblin@phys.ntu.edu.tw (G.-B.L.); ymchen@phys.ntu.edu.tw (Y.-M.C.); hhliu@phys.ntu.edu.tw (H.-H.L.); tsyr8924503@phys.ntu.edu.tw (F.-T.H.); a13956@phys.ntu.edu.tw (Y.K.); 2Biomedical & Molecular Imaging Center, National Taiwan University, Taipei 106319, Taiwan; 3Graduate Institute of Applied Physics, Biophysics Division, National Taiwan University, Taipei 106319, Taiwan

**Keywords:** rotenone, Parkinson’s disease, hyperthermia, neuroprotection, antioxidation

## Abstract

Parkinson’s disease (PD) is the second most prevalent neurodegenerative disease. It is characterized by mitochondrial dysfunction, increased reactive oxygen species (ROS), α-synuclein (α-syn) and phosphorylated-tau protein (p-tau) aggregation, and dopaminergic neuron cell death. Current drug therapies only provide temporary symptomatic relief and fail to stop or reverse disease progression due to the severe side effects or the blood–brain barrier. This study aimed to investigate the neuroprotective effects of an intermittent heating approach, thermal cycling-hyperthermia (TC-HT), in an in vitro PD model using rotenone (ROT)-induced human neural SH-SY5Y cells. Our results revealed that TC-HT pretreatment conferred neuroprotective effects in the ROT-induced in vitro PD model using human SH-SY5Y neuronal cells, including reducing ROT-induced mitochondrial apoptosis and ROS accumulation in SH-SY5Y cells. In addition, TC-HT also inhibited the expression of α-syn and p-tau through heat-activated pathways associated with sirtuin 1 (SIRT1) and heat-shock protein 70 (Hsp70), involved in protein chaperoning, and resulted in the phosphorylation of Akt and glycogen synthase kinase-3β (GSK-3β), which inhibit p-tau formation. These findings underscore the potential of TC-HT as an effective treatment for PD in vitro, supporting its further investigation in in vivo models with focused ultrasound (FUS) as a feasible heat-delivery approach.

## 1. Introduction

Parkinson’s disease (PD) is the second most prevalent neurodegenerative disease (NDD) after Alzheimer’s disease (AD) in elderly individuals and affects more than 8.5 million patients worldwide in 2019 [1,2]. PD is primarily characterized by the degeneration of dopaminergic neurons in the substantia nigra pars compacta, along with the abnormal accumulation of Lewy bodies in the striatum, resulting in clinical motor symptoms such as tremors, rigidity, akinesia, and postural instability [3]. Besides aging, exposure to pesticides such as rotenone (ROT) can also induce PD-like symptoms [4,5]. ROT, a natural compound derived from plants such as Derris and Lonchocarpus, has been utilized as a vegetable pest control agent for over a century [6]. It specifically inhibits mitochondrial complex I, which causes mitochondrial dysfunction and increases reactive oxygen species (ROS) production [7,8]. Furthermore, ROS promotes the aggregation of α-synuclein (α-syn) and phosphorylated-tau protein (p-tau), ultimately leading to neuronal death [4]. In a Wistar rat model exposed to 2 mg/kg ROT, motor symptoms associated with PD are observed, along with dopaminergic neurodegeneration in the substantia nigra, with Lewy body accumulation occurring in these neurons [5]. Hence, ROT-exposed cell and animal models serve as valuable models for research on PD therapy.

The mechanisms that trigger PD are currently under investigation. Current therapeutic approaches primarily focus on regulating dopamine and acetylcholine levels in the dopaminergic system of the striatum to alleviate associative motor symptoms [9]. These strategies include the use of levodopa to augment dopamine levels, monoamine oxidase-B inhibitors to prevent dopamine breakdown, and anticholinergic agents to reduce acetylcholine levels [10]. However, the effectiveness of drug treatment is often limited by the selective permeability of the blood–brain barrier or possible adverse effects, especially in elderly patients [10,11]. While these treatments provide temporary symptomatic relief, they do not address the underlying cause of PD, which could be attributed to the accumulation of misfolded proteins in neurons [12,13]. As a result, these treatments slow but do not halt or reverse disease progression. Therefore, it is crucial to find alternative treatments that target aggregated proteins to cure the disease.

Heat therapy has long been recognized as an effective alternative treatment, with hyperthermia (HT) resulting in therapeutic effects across various clinical fields, including ophthalmology, cardiology, and oncology [14]. Additionally, HT-induced moderate thermal stress can activate neuroprotective mechanisms that defend against oxidative stress and inhibit protein aggregation [15]. This is achieved by up-regulating enzymes involved in degradation [16], activating heat-shock protein 70 (Hsp70) for protein chaperoning [15,17], and increasing phosphorylated Akt (p-Akt) levels [18], which inactivate glycogen synthase kinase-3β (GSK-3β), thereby suppressing p-tau formation [19,20]. Moreover, the application of focused ultrasound (FUS) for heat delivery has additional advantages for NDD treatment, including the ability to penetrate the blood–brain barrier and precisely elevate the temperature in targeted brain regions [21,22], thereby mitigating the risks associated with invasive surgical procedures.

While heat therapy has therapeutic potential, caution is essential during its continuous application due to potential adverse effects. Studies have shown that continuous exposure to high temperatures may impair neural cells and potentially damage the central nervous system [23,24]. The continuous application of conventional HT treatment is associated with risks such as neurological and systemic complications, such as metabolic uncoupling and autoregulation loss, brain edema, intracranial hypertension, and intracerebral hemorrhage, leading to potential irreversible neural damage [25]. Therefore, it is crucial to develop a safer, more controllable, and more effective method for heat application during NDD treatment. Our previous research introduced a mild and intermittent thermal treatment, thermal-cycling hyperthermia (TC-HT), which contains repetitive low-temperature stages of minute-scale duration and enhances protective mechanisms in SH-SY5Y cells against oxidative stressors, such as hydrogen peroxide and β-amyloid (Aβ), while preserving cell viability [26]. Moreover, TC-HT has been demonstrated to reverse Aβ-induced cognitive and memory impairments in AD mouse models, while concurrently activating neuroprotective mechanisms in the hippocampus of these mice [27]. These findings suggest that TC-HT could be a potential therapeutic strategy for PD, as it might activate protective mechanisms in neuron cells without inducing adverse effects, and this strategy could be implemented into PD brains via programmable FUS to control the temperature and duration during treatment.

In this study, we used an ultrasound (US) exposure system to administer TC-HT treatment to SH-SY5Y cells, a human neuroblastoma cell line exhibiting dopaminergic properties and widely recognized as an in vitro model for PD research [28], aiming to investigate its therapeutic potential in a ROT-induced PD cell model. Our experimental design included assessments of mitochondrial apoptosis, intracellular ROS accumulation, and the expression of PD marker proteins such as α-synuclein and p-tau. In addition, we examined the involvement of neuroprotective mechanisms, including the sirtuin 1 (SIRT1)/Hsp70 and the Akt/GSK-3β signaling pathways, to explore the regulatory effects of TC-HT. These investigations offer mechanistic insights into the potential therapeutic role of US-based TC-HT in an in vitro ROT-induced PD model and provide a rationale for its further evaluation in in vivo models employing FUS as a heat-delivery approach.

## 2. Results

### 2.1. TC-HT Restores the ROT-Impaired Viability of SH-SY5Y Cells

In this study, we first evaluated ROT cytotoxicity in SH-SY5Y cells in a dose-dependent manner using the CCK-8 assay. As shown in Figure 1A, treatment with 5 μM of ROT reduces SH-SY5Y cell viability to 65.3% compared to untreated controls, which provides an appropriate range to evaluate the protective effects of the intermittent thermal treatment TC-HT and the continuous thermal treatment HT. Therefore, this concentration was subsequently used for further experiments. To determine the optimal temperature that could maximize heat-associated neuroprotective effects, SH-SY5Y cells were subjected to TC-HT at varying US intensities. Figure 1B indicates that a 9-cycle TC-HT exhibits no protective effect at US intensities below 1.0 W (*p* = 0.7492). At an intensity of 1.0 W, where the high temperature reached 40.0 ± 0.5 °C during the whole treatment process, TC-HT significantly restored the 5 μM ROT-impaired cell viability from 67.2% to 90.4% (*p* = 0.0044). However, at an intensity of 1.5 W, the recovery effect decreased to 77.3% with no statistical significance (*p* = 0.2832). Furthermore, we examine the neurotoxicity of TC-HT and HT at various intensities used alone. Our data also showed that 9-cycle TC-HT at a 1.5 W intensity exhibited significant neurotoxicity compared to untreated controls (75.9%, *p* = 0.0001), while 0.5 W (94.0%, *p* = 0.6369) and 1.0 W (92.8%, *p* = 0.4389) intensities did not significantly affect cell viability. In contrast, continuous HT for equal heat durations of 1.5 h as the 9-cycle TC-HT, also induced significant neurotoxicity at both 1.0 W (85.8%, *p* = 0.0169) and 1.5 W (43.6%, *p* < 0.0001) intensities but not at a 0.5 W intensity (88.5%, *p* = 0.0677) (Appendix A). These results suggest that the optimal US intensity for rescuing ROT-injured SH-SY5Y cells was 1.0 W in this research, corresponding to a stabilized high temperature of 40.0 ± 0.5 °C. Additionally, our previous study demonstrated that the total duration of heat application influences the neuroprotective efficacy of TC-HT [26]. Thus, TC-HT at a US intensity of 1.0 W was applied to SH-SY5Y cells for 6 and 9 cycles, and the results obtained under these conditions were compared to those obtained with continuous HT for equal heating durations of 1 and 1.5 h to assess the neuroprotective effects of TC-HT and HT. As shown in Figure 1C, 9-cycle TC-HT significantly improved the 5 μM ROT-reduced SH-SY5Y cell viability from 68.5% to 85.0% (*p* = 0.0005), outperforming the 81.0% recovery observed with 6-cycle TC-HT. In contrast, both 1 and 1.5 h HT did not result in significant protective effects on the 5 μM ROT-reduced cell viability (from 68.5% to 73.2% and 71.6%, respectively), indicating that continuous thermal treatment may be detrimental to neural cells. Moreover, there is a significant difference between the recovery effect of 9-cycle TC-HT and 1.5 h HT (*p* = 0.0031), but not between that of 6-cycle TC-HT and 1 h HT (*p* = 0.1031).

As a result, a 9-cycle TC-HT and a 1.5 h HT were chosen as the TC-HT and HT treatments for the subsequent experiments. In addition, a light microscope was utilized to assess the morphology of SH-SY5Y cells, as shown in Figure 1D, which indicated that TC-HT treatment maintained the structural integrity of ROT-impaired cells, while continuous HT did not.

### 2.2. TC-HT Attenuates ROT-Induced Mitochondrial Apoptosis

To evaluate the protective effects of TC-HT, we employed flow cytometry using the fluorescence dyes Annexin V-FITC and propidium iodide (PI) for apoptosis assessment, along with 5,5′,6,6′-Tetrachloro-1,1′,3,3′-tetraethylbenzimidazolylcarbocyanine chloride (JC-1) fluorescence staining, to assess the mitochondrial membrane potential (MMP). As illustrated in Figure 2A, TC-HT significantly reduced the apoptotic rates in the 5 μM ROT-exposed neural cells from 30.7% to 14.2% (*p* = 0.0015), greatly outperforming continuous HT with statistical significance, which only reduced the apoptotic rate to 25.7% (*p* = 0.0133). Additionally, the results of JC-1 staining in Figure 2B demonstrate that TC-HT significantly inhibited the 5 μM ROT-induced MMP loss, with the MMP level (normalized fluorescence intensity ratio of PE/FL1 channel) recovering from 0.65-fold to 0.85-fold of the control group (*p* < 0.0001), whereas continuous HT resulted in a slight and non-significant increase in the fluorescence intensity ratio to 0.67-fold of the control group (*p* = 0.7611), which is significantly lower than the recovering effect of TC-HT (*p* = 0.0001). Figure 2C presents confocal microscopy images of SH-SY5Y cells 24 h after each treatment, with red/green fluorescence intensities consistent with the results shown in Figure 2B.

### 2.3. TC-HT Reduces ROT-Induced Intracellular ROS and Promotes Antioxidative Protein Expression

To investigate the inhibitory effect of TC-HT on intracellular ROS levels, we performed flow cytometry to assess intracellular ROS levels in SH-SY5Y cells stained with the fluorescent dye DHE. As shown in Figure 3A, TC-HT significantly down-regulated the 5 μM ROT-induced increase in intracellular ROS levels in SH-SY5Y cells, decreasing it from 1.32-fold to 1.04-fold compared to the control (*p* = 0.0003). In contrast, continuous HT only slightly reduced ROS levels to 1.28-fold of the control, without any statistical significance (*p* = 0.7130), and the difference between TC-HT and continuous HT was statistically significant (*p* = 0.0009). Furthermore, as shown in Figure 3B, we examined the expression of the antioxidative enzyme superoxide dismutase 2 (SOD2), which reduced intracellular ROS levels by converting harmful O_2_^•−^ into less toxic hydrogen peroxide. Thus, TC-HT significantly restored the 5 μM ROT-reduced SOD2 expression from 0.81-fold to 1.18-fold of the control (*p* = 0.0013), surpassing the 0.76-fold expression observed with continuous HT (*p* = 0.0005), which showed no significant recovery effect (*p* = 0.8203).

### 2.4. TC-HT Down-Regulates the Expression of ROT-Induced PD Marker Proteins

To determine the expression levels of PD marker proteins, we used the AT8 antibody to recognize tau proteins phosphorylated at the serine 202 and threonine 205 residues. Our findings show that the 5 μM ROT treatment significantly increased the expression levels of α-syn and AT8 in SH-SY5Y cells to 1.25-fold and 1.47-fold of the control, respectively (Figure 4). TC-HT effectively reduced the expression level of α-syn to 1.07-fold and AT8 to 1.15-fold of the control, with both reductions achieving statistical significance (*p* = 0.0288 and 0.0022, respectively). In contrast, continuous HT did not significantly affect the expression levels of α-syn (1.28-fold, *p* = 0.8756) and AT8 (1.40-fold, *p* = 0.6245) compared to the 5 μM ROT treatment, and both marker proteins showed significantly lower expression levels under TC-HT compared to HT (*p* = 0.0106 and 0.0097, respectively).

### 2.5. TC-HT Attenuates ROT-Induced PD In Vitro via Heat-Activated Mechanisms

Given that TC-HT effectively down-regulates ROT-induced PD markers in SH-SY5Y cells, we further investigated the expression of neuroprotective proteins, particularly focusing on the inhibition of α-syn and p-tau accumulation. Our findings show that both Hsp70 and its upstream transcription factor, SIRT1 [29], were significantly up-regulated following TC-HT treatment, with Hsp70 levels increasing from 0.70-fold to 0.90-fold (*p* = 0.0002) and SIRT1 from 0.67-fold to 0.83-fold (*p* = 0.0026) of the control (Figure 5A,B), surpassing the expression levels of 0.65-fold (*p* = 0.0003) and 0.53-fold (*p* = 0.0002) observed with continuous HT. Moreover, HT did not significantly affect Hsp70 levels compared to 5 μM ROT treatment alone (*p* = 0.1892). In addition, Figure 5C,D shows that the 5 μM ROT treatment down-regulates p-Akt and p-GSK-3β (Ser9), while TC-HT effectively reverses this decline, significantly improving p-Akt level from 0.75-fold to 1.25-fold (*p* < 0.0001) and p-GSK-3β (Ser9) level from 0.66-fold to 0.88-fold of the control (*p* = 0.0142). These results significantly surpass the expression levels of 0.88-fold for p-Akt and 0.60-fold for p-GSK-3β (Ser9) observed with continuous HT (*p* = 0.0001 and 0.0030, respectively), which do not show statistical significance compared to the 5 μM ROT-treated SH-SY5Y cells (*p* = 0.0536 and 0.6163 for p-Akt and p-GSK-3β (Ser9), respectively).

## 3. Discussion

As the second most common NDD, PD is characterized by the loss of dopaminergic neurons in the substantia nigra pars compacta, accompanied by the extensive accumulation of Lewy bodies and tau fibrils, which consist of aggregated α-syn and p-tau proteins [12,13,30]. Current primary treatments for PD are pharmacological, but they provide only temporary symptomatic relief [10] and do not address the underlying issue of protein aggregation in dysfunctional dopaminergic neurons, making it challenging to halt PD progression. Therefore, heat application has emerged as a promising alternative treatment, as it can potentially enhance neuroprotective effects and bypass the challenges associated with crossing the blood–brain barrier, enabling the direct targeting of affected brain regions. In this study, we employ a modified heat treatment method, TC-HT, which includes an intermittent low temperature stage of minute-scale duration and offers enhanced safety and superior neuroprotective effects compared to continuous HT, and investigate its therapeutic potential in a ROT-induced in vitro PD model.

The accumulation of ROT, a widely used pesticide, can occur in humans through the food chain. Studies have shown that ROT induces dopaminergic neuron degeneration and Lewy body accumulation in the substantia nigra pars compacta in mice [5], leading to PD-like motor dysfunction. Thus, ROT has become a widely used PD inducer in both in vitro and in vivo studies. In this study, 5 μM of ROT was employed to impair the dopaminergic-like neuroblastoma cell line SH-SY5Y, which was also used in other research [31,32] and provides a suitable range to investigate the neuroprotective effects and mechanisms of TC-HT. Notably, the heat delivery method as well as the neurotoxic agent used in the study are different from our previous study [26]. These differences required re-optimization of TC-HT parameters to ensure effective neuroprotection in the study. This was particularly important for the heating temperature and duration of heat exposure, as excessive heating could damage cells, while insufficient heating would fail to adequately activate neuroprotective mechanisms. The findings of this study indicate that TC-HT, utilizing a US intensity of 1W and maintaining a maximum temperature of 40 ± 0.5 °C over a total of nine cycles, significantly preserves the viability of SH-SY5Y cells exposed to ROT more effectively than continuous HT, as illustrated in Figure 1. This underscores the enhanced neuroprotective effects of TC-HT against ROT in comparison to continuous HT.

Apoptosis, closely related to mitochondrial function, is a key mechanism of cell death. Mitochondria are crucial for providing cellular energy and maintaining cell viability. Studies have shown that ROT exposure could induce mitochondrial dysfunction, leading to cell death [5]. However, it was found that moderate thermal exposure could restore the impaired MMP [26]. Our results also demonstrate that TC-HT significantly reduces apoptosis in ROT-injured SH-SY5Y cells, with superior efficacy compared to continuous HT (Figure 2A). Additionally, the JC-1 staining results in Figure 2B,C suggest that TC-HT restores MMP more effectively in ROT-treated SH-SY5Y cells than continuous HT, indicating that TC-HT has a better protective effect on mitochondrial integrity against ROT-induced damage. These results collectively highlight the superior ability of TC-HT to reduce ROT-induced mitochondrial apoptosis compared to continuous HT application in SH-SY5Y cells.

In addition to inducing apoptosis, impaired mitochondrial function also results in excessive ROS production, which is detrimental to neural cells in PD and implicated in the progression of NDDs [33]. Moreover, dopamine metabolism results in ROS production in the neurons of the substantia nigra pars compacta, making these neurons particularly susceptible to oxidative stress [7]. Hence, controlling ROS levels is important for neuronal survival in PD. Studies have shown that heat and ultrasound stimulations can alleviate intracellular oxidative stress by activating the antioxidative mechanisms [26,34,35]. As shown in Figure 3A, our study demonstrates that TC-HT effectively down-regulates ROT-elevated ROS levels in SH-SY5Y cells, resulting in significantly higher antioxidative effects compared to continuous HT. Notably, this reduction is accompanied by a significant increase in the expression of the antioxidative enzyme SOD2 (Figure 3B). Therefore, TC-HT can regulate intracellular ROS by up-regulating SOD2 and thereby mitigating ROT-induced injury in SH-SY5Y cells. In addition to SOD2, it is noteworthy that thermal stimulation may trigger various antioxidative responses. A recent study has demonstrated that mild heat shock at 40 °C can activate the nuclear factor E2-related factor 2/antioxidant response element (Nrf2/ARE) signaling pathway, promoting antioxidant responses beyond SOD2. In particular, this pathway has been shown to increase the expression of autophagy-related proteins, thereby enhancing cellular stress tolerance [36]. Our previous study has also shown that TC-HT can increase Nrf2 expression in H_2_O_2_-treated SH-SY5Y cell models [26]. These findings suggest that TC-HT may engage broader antioxidative and neuroprotective mechanisms, which warrants further investigation.

The accumulation of α-syn and p-tau, which aggregate as Lewy bodies and tau fibrils in the degenerated dopaminergic neurons of the substantia nigra, are critical pathological hallmarks of PD [12,13,30] and contribute to damaged mitochondria and excessive ROS production, which results in mitochondrial dysfunction and dopaminergic neuron loss [30,37,38]. ROS overproduction can further promote the aggregation of α-syn and p-tau proteins, exacerbating PD progression [30,39,40]. Thus, the regulation of α-syn and p-tau expression levels is critical for protecting neural cells in PD. As shown in Figure 4, TC-HT significantly reduces the expression levels of α-syn and AT8 in ROT-damaged SH-SY5Y cells. This reduction is significantly larger than that observed with continuous HT and aligns with the observed inhibitory effects on mitochondrial apoptosis (Figure 2) and intracellular ROS levels (Figure 3). Hence, TC-HT demonstrates considerable potential as a therapeutic approach for PD.

The expression levels of aggregated α-syn and p-tau can be modulated through various mechanisms, with chaperoning playing a key role in preventing their accumulation. Hsp70, an important heat-activated chaperone protein, helps to manage misfolded proteins such as α-syn and p-tau, protecting neuronal cells from PD marker accumulation and facilitating their degradation [41,42]. Our results have shown that TC-HT significantly increases the expression levels of Hsp70 and its upstream transcriptional factor SIRT1 (Figure 5A,B), inducing stronger neuroprotective effects on ROT-impaired SH-SY5Y cells compared to continuous HT and alleviating the stress resulting from ROT-induced α-syn and p-tau accumulations more effectively. Additionally, SIRT1 also acts as an upstream regulator of SOD2 [43]. Accompanied by the results in Figure 2, these findings suggest that TC-HT results in stronger antioxidative effects than continuous HT against ROT-induced oxidative stress by up-regulating SIRT1 more effectively. Besides chaperoning, preventing the formation of aggregated proteins is crucial for inhibiting their pathological effects. GSK-3β, a protein involved in tau hyperphosphorylation, can be inactivated by heat-activated protein p-Akt through the phosphorylation of its Ser9 residue, which reduces p-tau formation [19,20]. As shown in Figure 5C,D, TC-HT significantly activates greater p-Akt levels compared to continuous HT, consequently leading to the inactivation of more GSK-3β through the increased expression of p-GSK-3β (Ser9). These results, along with the reduced AT8 levels in Figure 4B, suggest that TC-HT reduces p-tau protein accumulation by activating p-Akt, which inhibits p-tau formation by decreasing p-GSK-3β (Ser9) activation. Collectively, TC-HT reduces the expression levels of ROT-induced α-syn and p-tau more effectively than continuous HT by robustly enhancing the neuroprotective mechanisms associated with protein chaperoning and aggregation inhibition.

Physical stimulation is increasingly recognized as a potent therapeutic approach for NDDs. Instead of continuous HT application, intermittent thermal implementation can preserve neuroprotective effects while minimizing side effects [26,27]. On the other hand, it has been reported that a 40 Hz repetition frequency in both light and US treatments might help to treat AD in vivo by decreasing Aβ accumulation and attenuating neuroinflammation [44,45], with phase I clinical trials already underway [46]. Inspired by these findings, we integrated the 40 Hz US pulse as a repetition frequency into our TC-HT treatment for a ROT-induced in vitro PD model. Our results show that, while mono TC-HT treatment significantly improves cell viability (*p* = 0.0008), TC-HT with additional 40 Hz repetition US pulses also exhibits a protective effect on the viability of SH-SY5Y cells against 5 μM of ROT, which is, however, only slightly higher than that of mono TC-HT, and the difference is not statistically significant (*p* = 0.9853) (Figure 6). This suggests that further investigation of optimized pulse repetition frequencies in physical stimulations could enhance therapeutic efficacy, potentially expanding their applicability for treating NDDs.

Heat therapy can facilitate the rescue of impaired neural cells, but continuous heat exposure may result in adverse effects, including neural cell damage, neurological and systemic issues, or even harm to the central nervous system [23,24,25]. Our study shows that, with the same maximum temperature range of 40 ± 0.5 °C and total heat exposure time, TC-HT achieves significantly greater neuroprotective effects than continuous HT in the ROT-induced in vitro PD model. Although this study is conducted in an in vitro PD model, the results indicate that interspersing low-temperature intervals during heat stimulation periods is crucial for maximizing the neuroprotective effect of thermal stimulation. Importantly, the duration of these intermittent low-temperature intervals is a significant factor affecting the effectiveness of TC-HT. Different models necessitate varying durations to achieve the best results. For instance, a 35 s low-temperature duration produced the most favorable neuroprotective effects in a hydrogen peroxide-induced in vitro NDD model [26], while a 2 min duration resulted in the most effective viral clearance in a U-937 macrophage model [47]. Notably, both studies indicated that prolonging the low-temperature intervals beyond their optimal duration could negate the beneficial effects of treatment, as referenced in studies [26,47]. This suggests that the signaling pathways activated by heat may be disrupted if the intermittent low-temperature intervals are excessively extended. Therefore, careful adjustment of the low-temperature duration is vital for enhancing TC-HT effectiveness across various models.

The research has demonstrated the neuroprotective benefits of TC-HT in the ROT-induced in vitro PD model, and further in vivo studies are necessary to confirm the therapeutic effects of TC-HT on PD behavioral symptoms and the pathological accumulation of protein tangles and fibrils in PD brains. In addition, radiofrequency electromagnetic fields and FUS are potential practical modalities for delivering TC-HT to PD animals or patients with PD disease [21,22,48,49]. Particularly, FUS can provide programmable control and precise, localized heating in the human brain in a non-invasive manner [21,22]. Additionally, the heating area and temperature can be modulated via multi-focal FUS by adjusting the scanned spot size and intensity. This precise thermal regulation allows efficient heat dissipation and helps achieve the desired temperature for TC-HT, thereby enhancing its therapeutic potential. These findings highlight the therapeutic potential of FUS-based TC-HT, but several challenges must be addressed before clinical implementation, including ensuring precise and uniform thermal delivery to the targeted brain regions, managing patient-specific variability in thermal sensitivity and substantial tissue motion, and developing non-invasive systems for real-time temperature monitoring [50,51,52]. Magnetic resonance imaging-guided FUS systems with integrated thermometry may help overcome these limitations and facilitate the safe application of TC-HT in clinical settings [50,52,53]. While FUS-based TC-HT shows promising development, its broader application remains technically challenging. To further explore the therapeutic potential of FUS-based TC-HT and its practical limitations, future studies should establish protocols for FUS-based TC-HT in in vivo models, including optimizing the temperature settings and treatment durations, and explore its potential in combination with clinical PD drugs or other therapeutic strategies to support future human trials.

In summary, this study presents US-based TC-HT as a safer and more effective hyperthermic method for protecting SH-SY5Y cells in an in vitro ROT-induced PD model. US-based TC-HT demonstrates superior protective effects in preserving ROT-impaired cell viability and reducing apoptotic rates compared to US-based continuous HT. Additionally, TC-HT significantly reduces ROT-induced MMP loss and intracellular ROS accumulation, while up-regulating the antioxidative enzyme SOD2. In comparison, continuous HT shows only minimal effects on ROT-induced MMP loss and an increase in ROS levels. Furthermore, TC-HT significantly outperforms continuous HT in down-regulating PD marker proteins such as α-syn and p-tau. This enhanced efficacy could be attributed to the substantial increase in the levels of Hsp70 and its transcriptional regulator SIRT1, which together prevent the further accumulation of α-syn and p-tau through effective chaperoning. Moreover, TC-HT results in the stronger activation of Akt, leading to more effective inhibition of the hyperphosphorylation of GSK-3β and thus resulting in a greater reduction in ROT-induced p-tau expression (Figure 7). Collectively, our findings demonstrate the superior therapeutic potential of the US-based intermittent heating method TC-HT over the US-based continuous heating method HT for an ROT-induced PD model in vitro, highlighting its efficacy and safety as a hyperthermic approach. Practically, programmable FUS could serve as an optimal heating source to implement TC-HT as a treatment strategy for PD, while its practical limitation requires further exploration in in vivo models and potential human trials. Notably, although the heat delivery method and the neurotoxic agent used in the study differ from our previous study [26], TC-HT still demonstrated a significant neuroprotective effect. This suggests that, as a hyperthermic approach, the achieved temperature is a more critical factor of TC-HT neuroprotective efficacy than the specific delivery device or heating method. Therefore, future studies aiming to replicate this methodology should calibrate their systems to achieve comparable cell temperatures (40 ± 0.5 °C), rather than directly adopting the ultrasound settings.

## 4. Materials and Methods

### 4.1. Cell Culture

The human neuroblastoma cell line SH-SY5Y was purchased from American Type Culture Collection (Manassas, VA, USA) and was cultured in Dulbecco’s Modified Eagle Medium (DMEM) (HyClone; Cytiva, Marlborough, MA, USA) supplemented with 10% fetal bovine serum (FBS) (HyClone; Cytiva), 100 units/mL penicillin, and 100 µg/mL streptomycin (Gibco Life Technologies, Grand Island, NY, USA). The cells were maintained in a humidified incubator with 5% carbon dioxide (CO_2_) at a temperature of 37 °C. Once the confluence reached 80%, cells were subcultured with a 0.05% trypsin–0.5 mM ethylenediamine tetraacetic acid (EDTA) solution (Gibco Life Technologies), and subsequently seeded into 96-well or 35 mm diameter culture dishes (Thermo Fisher Scientific, Inc., Waltham, MA, USA) for in vitro experiments after a 24 h incubation in the humidified 5% CO_2_ incubator at 37 °C.

### 4.2. Ultrasound (US) Exposure

The US exposure system consisted of a function generator (SG382; Stanford Research Systems, Sunnyvale, CA, USA), a power amplifier (25a250a; Amplifier Research, Souderton, PA, USA), and a 0.5 MHz ceramic planar US transducer (A101S-RM; Olympus NDT Inc., Waltham, MA, USA), as shown in Figure 8. The function generator produced continuous pulses with the following parameters: −10 dBm amplitude, 1 ms pulse period, and 0.5 ms pulse width. The cell culture plate or dish was positioned on the US transducer, with glycerol filling between the gap to transmit US waves into the culture plate or dish. The ceramic planar US transducer facilitated the conversion of electrical signals into acoustic and thermal energy, thereby administering both TC-HT and HT treatments to the SH-SY5Y cells.

### 4.3. TC-HT Treatment

As described in our previous study [26,27], the intermittent heating method TC-HT is a repetitive procedure comprising multiple cycles, each consisting of a designated high-temperature stage, followed by a cooling stage to achieve a series of short periods of heat exposure over a specific duration. In the present study, the actual temperatures experienced by SH-SY5Y cells were monitored using a K-type thermocouple in conjunction with a temperature/process controller, and a 9-cycle repeated TC-HT was implemented. The high temperature stage was sustained for 10 min, during which the maximum temperature remained at 40 ± 0.5 °C with the US intensity set at 1.0 W, followed by a natural thermal dissipation period of 1 min (without US energy transfer) for cooling. For the HT application at the same US intensity, the temperature was continuously maintained at 40 ± 0.5 °C for a duration of 90 min, with the US intensity set at 1.0 W. The monitored actual temperatures during TC-HT or HT applications are depicted in Figure 9. Additionally, to evaluate the uniformity of heating, we conducted temperature measurements at various locations within the central region (2 cm diameter) of the planar ultrasound transducer (3 cm diameter), where the majority of cells were situated. As demonstrated in Appendix A, only minor temperature fluctuations were recorded across this area, indicating that the temperature variations resulting from the unevenness of the US field are negligible. In order to conduct TC-HT utilizing US pulses at a frequency of 40 Hz, the modulation mode of the function generator was set to pulse mode, characterized by a period of 25 ms and a pulse duty cycle of 50%. The treatment of SH-SY5Y cells was performed at the same temperature, maintaining the output US intensity at 1 W, with the maximum temperature recorded at 40 ± 0.5 °C.

### 4.4. Rotenone Treatment

The SH-SY5Y cells were pretreated with TC-HT or HT at room temperature, and subsequently maintained in the humidified 5% CO_2_ incubator at 37 °C. At 4 h after the thermal treatment, the pretreated SH-SY5Y cells were exposed to 5 µM of rotenone, which was purchased from Sigma-Aldrich (St. Louis, MO, USA) and dissolved in dimethyl sulfoxide (DMSO) (Echo Chemical Co., Ltd., Miaoli, Taiwan) at a stock concentration of 100 mM.

### 4.5. Cell Viability Assessment

Cell viability of the treated SH-SY5Y cells was evaluated using the Cell counting kit-8 (CCK-8) assay (Dojindo, Kumamoto, Japan). Specifically, 10 μL of CCK-8 solution was added to each well containing either treated or untreated SH-SY5Y cells at 24 h after ROT treatment, facilitating the formation of formazan from live cells. After a 3 h incubation in the 5% CO_2_ incubator at 37 °C, the absorbance of each well was quantified using a Multiskan GO microplate spectrophotometer (Thermo Fisher Scientific). The amount of formazan was determined by measuring the absorbance at 450 nm, with a background subtraction at 600 nm. Cell viability was expressed as a percentage, with the viability of untreated control cells designated as 100%.

### 4.6. Morphological Observation of SH-SY5Y Cells Using Light Microscopy

To observe the morphology changes of SH-SY5Y cells after treatment, an inverted light microscope (Leica DM IRB; Leica, Wetzlar, Germany) was used to capture light microscopy images.

### 4.7. Apoptotic Analysis via Flow Cytometry

The apoptotic rates of SH-SY5Y cells were assessed using Annexin V-Fluorescein isothiocyanate (Annexin V-FITC) and propidium iodide (PI) double detection kit (BD Biosciences, San Jose, CA, USA). At 24 h after ROT treatment, SH-SY5Y cells were harvested and rinsed twice with ice-cold phosphate-buffered saline (PBS) (Hyclone). The rinsed cells were then resuspended in a binding buffer containing Annexin V-FITC and PI, followed by a 15-min incubation at room temperature in the dark. Apoptotic signals were measured by the FACSCanto II system (BD Biosciences) with the PE and FL1 channels.

### 4.8. Detection of ROS Levels via Flow Cytometry

ROS levels of superoxide anion (O_2_^•−^) were measured using fluorescent dye dihydroethidium (DHE) (Sigma-Aldrich). At 24 h after ROT treatment, SH-SY5Y cells were harvested and rinsed twice with PBS. The rinsed cells were then resuspended in PBS containing 5 μM of DHE and incubated in the dark for 30 min at 37 °C. ROS signals were detected using the FACSCanto II system with the PE channel.

### 4.9. MMP Analysis via Flow Cytometry

The loss of MMP was evaluated using the mitochondrial fluorescent dye JC-1 (MedChemExpress Ltd., Monmouth Junction, NJ, USA). At 24 h after ROT treatment, SH-SY5Y cells were harvested and rinsed with PBS before staining. The rinsed cells were then resuspended in PBS containing 5 μg/mL JC-1 and incubated in the dark for 30 min at 37 °C. The fluorescent signals from JC-1 were measured using the FACSCanto II system with the PE channel and FL1 channels. The levels of MMP were determined by calculating the normalized fluorescence intensity ratio of PE/FL1 channel relative to the untreated control cells.

### 4.10. JC-1 Fluorescence Detection by Laser Scanning Confocal Microscopy

In order to observe fluorescent signals through microscopy, SH-SY5Y cells were subjected to staining with JC-1 at a concentration of 5 μg/mL and incubated in the dark for 30 min at 37 °C. A Zeiss LSM 880 inverted laser scanning confocal microscope (Carl Zeiss GmbH, Jena, Germany) was employed to excite the red and green fluorescence of JC-1 at wavelengths of 565 nm and 488 nm, respectively, facilitating the capture of fluorescent images.

### 4.11. Western Blot Analysis

The protein expression levels in SH-SY5Y cells were analyzed using Western blot analysis. Following a wash with PBS, the cells were lysed in a lysis buffer (Millipore, Billerica, MA, USA) containing 50 mM of Tris-HCl (pH 7.4) and 0.15 M of NaCl, which was supplemented with an active protease/phosphatase inhibitor cocktail (#5827S; Cell Signaling Technology, Danvers, MA, USA). After centrifugation at 23,000× *g* for 30 min at 4 °C, the supernatants were collected, and protein concentrations were quantified using the Bradford protein assay (Bioshop, Inc., Burlington, ON, Canada). Subsequently, 20 μg of protein from each sample was separated by 10% SDS-polyacrylamide gel electrophoresis (SDS-PAGE) and transferred onto polyvinylidene fluoride (PVDF) membranes (Millipore). To block non-specific antibody binding, the membranes were incubated with 5% bovine serum albumin in TBST (20 mM Tris-base, pH 7.6, 0.15 M NaCl, 0.1% Tween 20 (Bioshop, Inc.)) for 1 h at room temperature. Primary antibodies targeting SOD2 (1:1000 dilution), Hsp70 (1:1000 dilution), SIRT1 (1:1000 dilution), p-Akt (1:1000 dilution) (Cell Signaling), α-syn (1:300 dilution), AT8 (1:1000 dilution) (p-tau), phosphorylated GSK-3β (serine 9) (p-GSK-3β (Ser9); 1:1000 dilution) (Abcam, Cambridge, UK), and GAPDH (1:10,000 dilution) (Gentex, Irvine, CA, USA) in blocking solution were subsequently applied to the membranes for probing and incubated overnight at 4 °C. Following three washes with TBST, the membranes were exposed to horseradish peroxidase-conjugated goat anti-rabbit secondary antibody (1:10,000 dilution) (Jackson ImmunoResearch Laboratories, West Grove, PA, USA) in blocking solution at room temperature for 1 h. Immunoreactivity signals were amplified by an enhanced chemiluminescence (ECL) substrate (Advansta, San Jose, CA, USA) and visualized using an imaging system (Amersham Imager 600, GE Healthcare Life Sciences, Marlborough, MA, USA). Images were analyzed with Image Lab software (version 6.0.1, Bio-Rad Laboratories, Inc., Hercules, CA, USA). GAPDH was utilized as a loading control to normalize the relative expression levels of target proteins.

### 4.12. Statistical Analysis

All experiments were conducted in triplicate to ensure validation, and results are expressed as mean ± standard deviation. Statistical analyses were performed using one-way analysis of variance (ANOVA) with OriginPro 2015 software (version 92E; OriginLab Corporation, Northampton, MA, USA), and results were deemed statistically significant when *p*-values were less than 0.05.

## Figures and Tables

**Figure 1 ijms-26-06671-f001:**
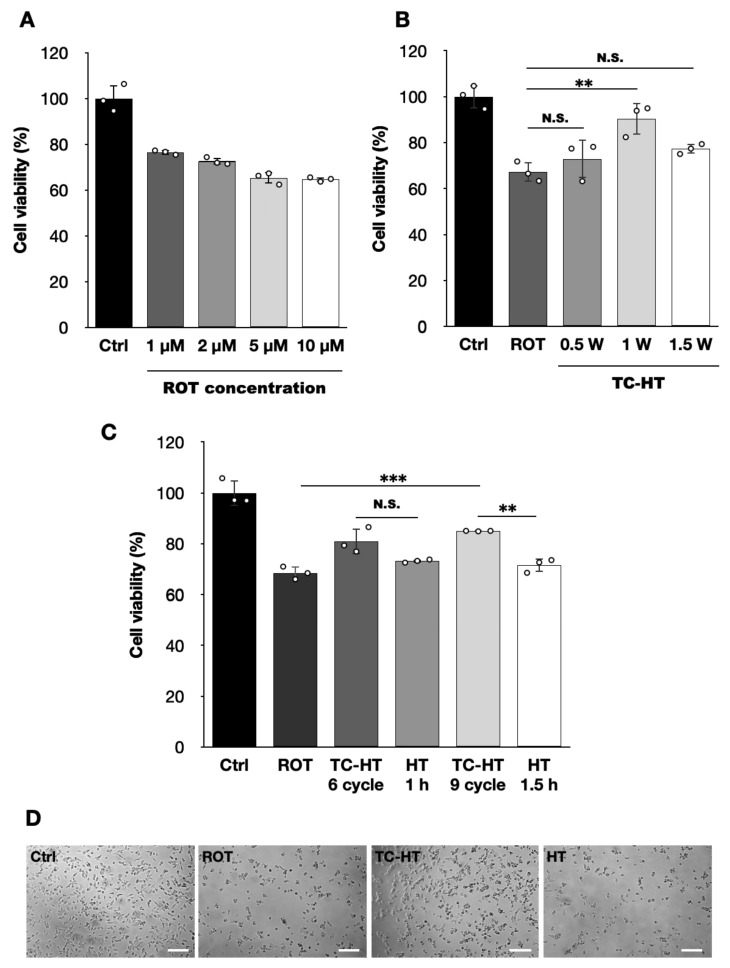
Neuroprotective effects of TC-HT on SH-SY5Y against ROT cytotoxicity. CCK-8 assay was conducted to determine the viability of SH-SY5Y cells following (**A**) different ROT concentrations (*F*-value = 66.6585), (**B**) TC-HT treatments with different US intensities (*F*-value = 16.0402), and (**C**) TC-HT treatments with different cycles, along with corresponding HT treatments with equal heating durations (*F*-value = 38.0758). (**D**) Representative SH-SY5Y cell morphology after treatment was imaged using a light microscope. Scale bar = 100 μm. All TC-HT and HT treatments in (**B**,**C**) were performed at 5 μM of ROT. The viabilities were measured 24 h after ROT treatment. Data are presented as the mean ± standard deviation in triplicate. Significance levels between indicated groups are denoted as ** *p* < 0.01 and *** *p* < 0.001, while non-significant differences are indicated as N.S.

**Figure 2 ijms-26-06671-f002:**
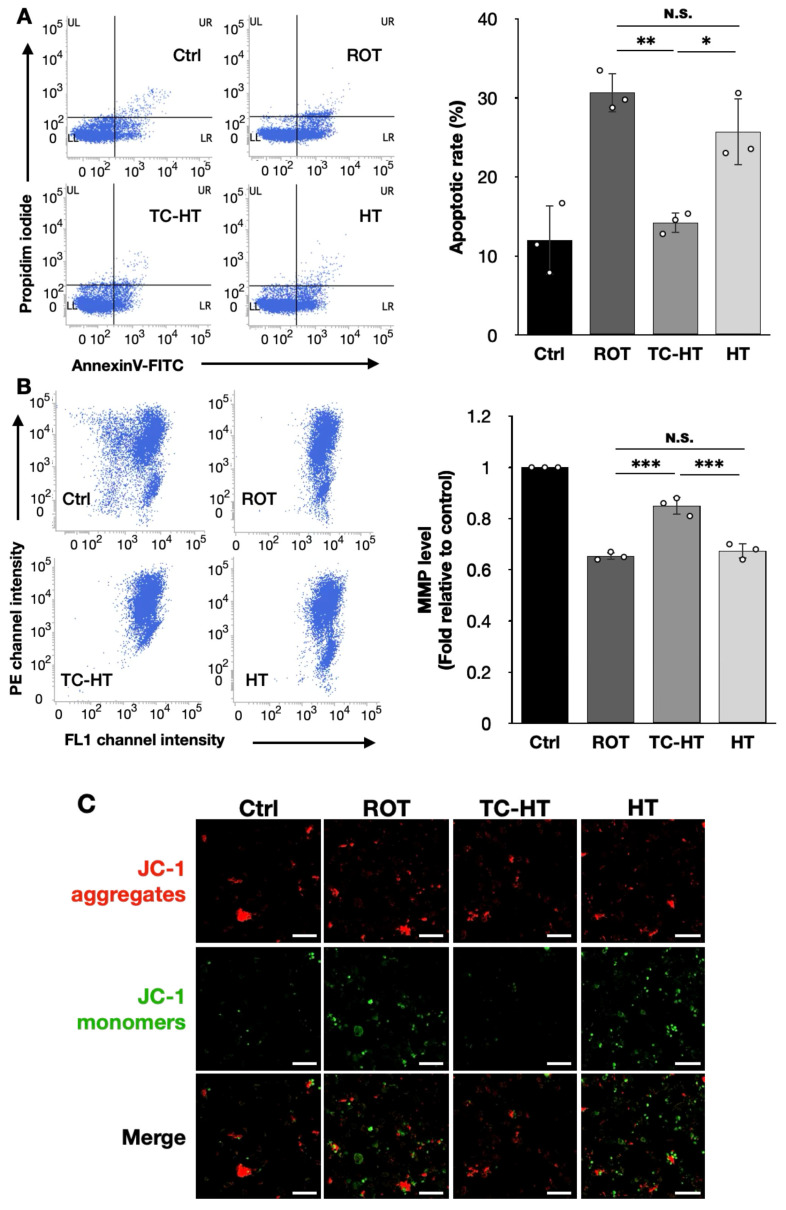
TC-HT attenuates ROT-induced mitochondrial apoptosis. (**A**) Apoptosis after treatment was analyzed using flow cytometry using Annexin V-FITC/PI double staining, and the apoptotic percentage (LR+UR) was calculated (*F*-value = 21.3397). (**B**) MMP level after treatment was analyzed via flow cytometry with JC-1 fluorescence dye, and the MMP levels after each treatment were determined by the normalized fluorescence intensity ratio of PE/FL1 channel relative to control (*F*-value = 129.5991). (**C**) The red and green fluorescence signals of JC-1 after treatment were imaged by a confocal microscope. Scale bar = 100 µm. All TC-HT and HT treatments were performed at 5 μM of ROT. Data are presented as the mean ± standard deviation in triplicate. Significance levels between indicated groups are denoted as * *p* < 0.05, ** *p* < 0.01, and *** *p* < 0.001, while non-significant differences are indicated as N.S.

**Figure 3 ijms-26-06671-f003:**
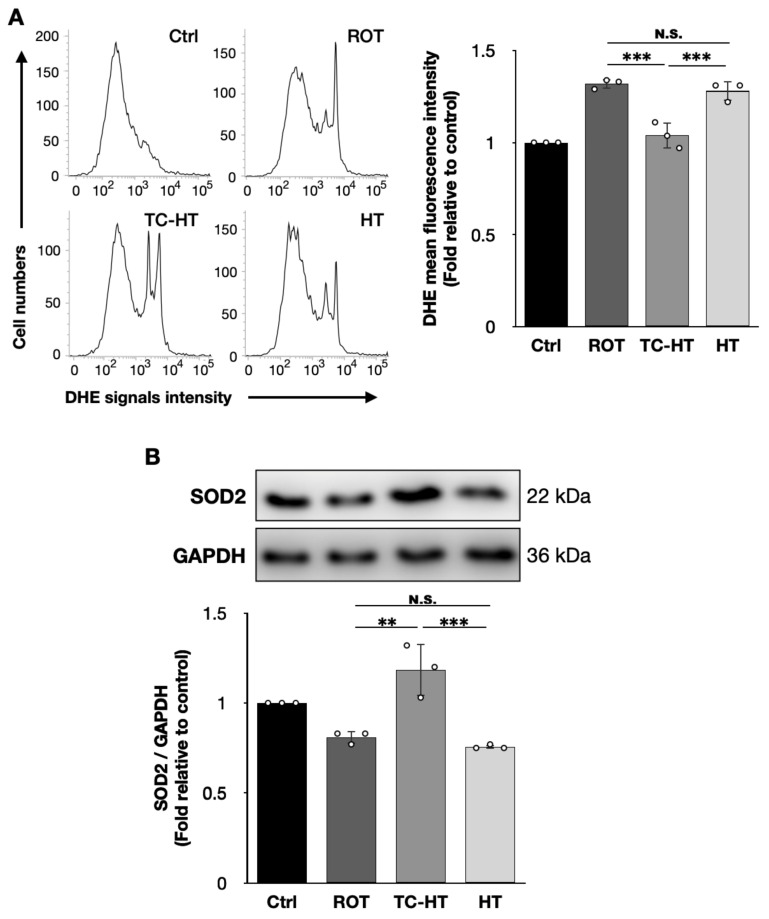
TC-HT reduces ROT-induced intracellular ROS via SOD2 promotion. (**A**) Intracellular superoxide radical anion (O_2_^•−^) levels of SH-SY5Y cells were determined by flow cytometry using the fluorescent dye DHE, and the DHE mean fluorescence intensities after each treatment were calculated (*F*-value = 38.5542). (**B**) Representative Western blots of the antioxidative enzyme SOD2 and its expression levels after each treatment were quantified (*F*-value = 20.0822). Glyceraldehyde-3-phosphate dehydrogenase (GAPDH) was used as a loading control. All TC-HT and HT treatments were performed at 5 μM of ROT. Data are presented as the mean ± standard deviation in triplicate. Significance levels between indicated groups are denoted as ** *p* < 0.01 and *** *p* < 0.001, while non-significant differences are indicated as N.S.

**Figure 4 ijms-26-06671-f004:**
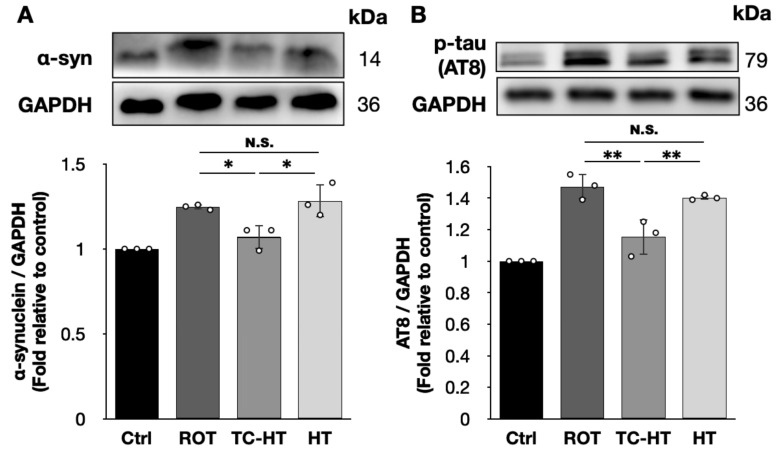
TC-HT down-regulates the expression levels of ROT-induced PD marker proteins. Representative Western blots and the quantifications of (**A**) α-syn (*F*-value = 15.4900) and (**B**) AT8 (*F*-value = 30.0535) after each treatment. GAPDH was used as a loading control. All TC-HT and HT treatments were performed at 5 μM of ROT. Data are presented as the mean ± standard deviation in triplicate. Significance levels between indicated groups are denoted as * *p* < 0.05 and ** *p* < 0.01, while non-significant differences are indicated as N.S.

**Figure 5 ijms-26-06671-f005:**
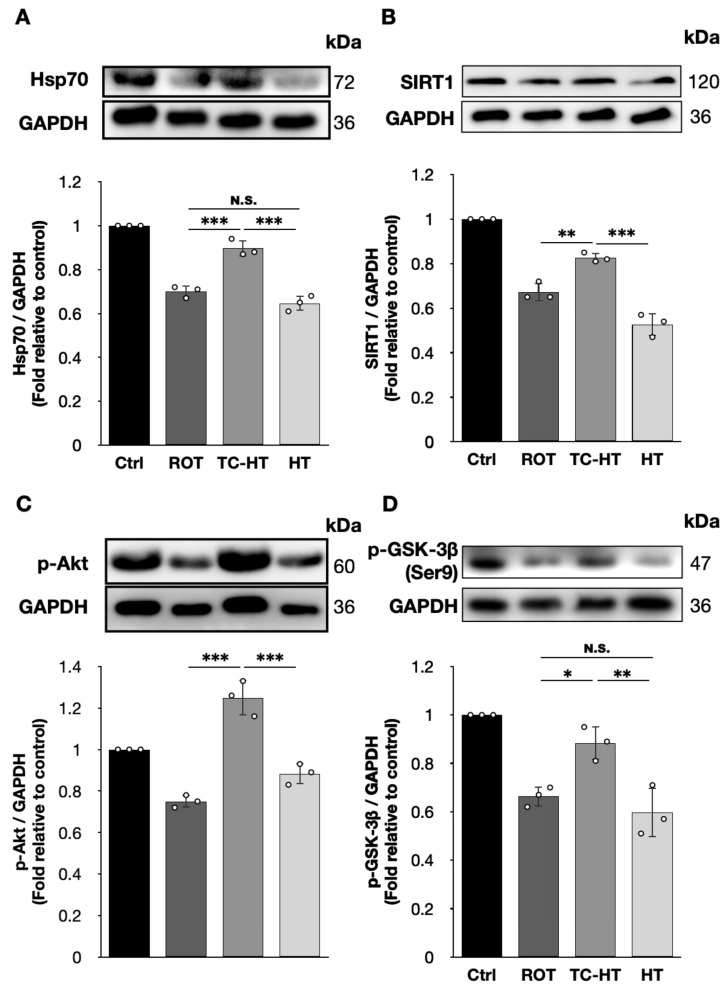
TC-HT promotes the neuroprotective mechanisms against the accumulation of PD marker proteins. Representative Western blots and the quantifications of (**A**) Hsp70 (*F*-value = 97.8845), (**B**) SIRT1 (*F*-value = 105.4941), (**C**) p-Akt (*F*-value = 50.3882), and (**D**) p-GSK-3β (Ser9) (*F*-value = 24.8337) after each treatment. GAPDH was used as a loading control. All TC-HT and HT treatments were performed at 5 μM of ROT. Data are presented as the mean ± standard deviation in triplicate. Significance levels between indicated groups are denoted as * *p* < 0.05, ** *p* < 0.01, *** *p* < 0.001, while non-significant differences are indicated as N.S.

**Figure 6 ijms-26-06671-f006:**
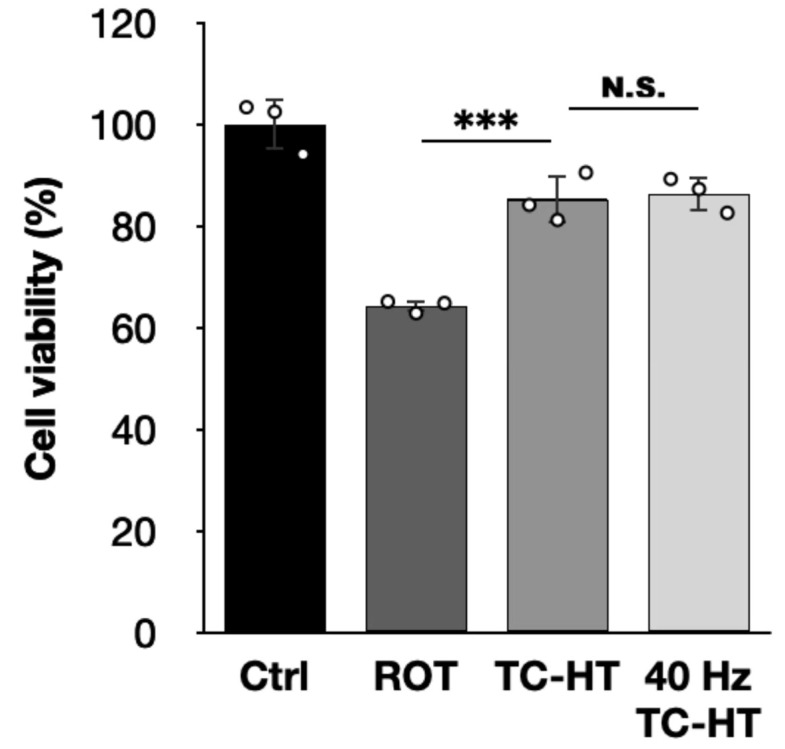
The neuroprotective effect of TC-HT integrated with additional 40 Hz repetition US pulses against ROT cytotoxicity. CCK-8 assay was conducted to determine the viability of SH-SY5Y cells following mono TC-HT or TC-HT with 40 Hz repetition US pulses. The viabilities were measured 24 h after ROT treatment (*F*-value = 41.7929). All TC-HT and HT treatments were performed at 5 μM of ROT. Data are presented as the mean ± standard deviation in triplicate. Significance levels between indicated groups are denoted as *** *p* < 0.001, while non-significant differences are indicated as N.S.

**Figure 7 ijms-26-06671-f007:**
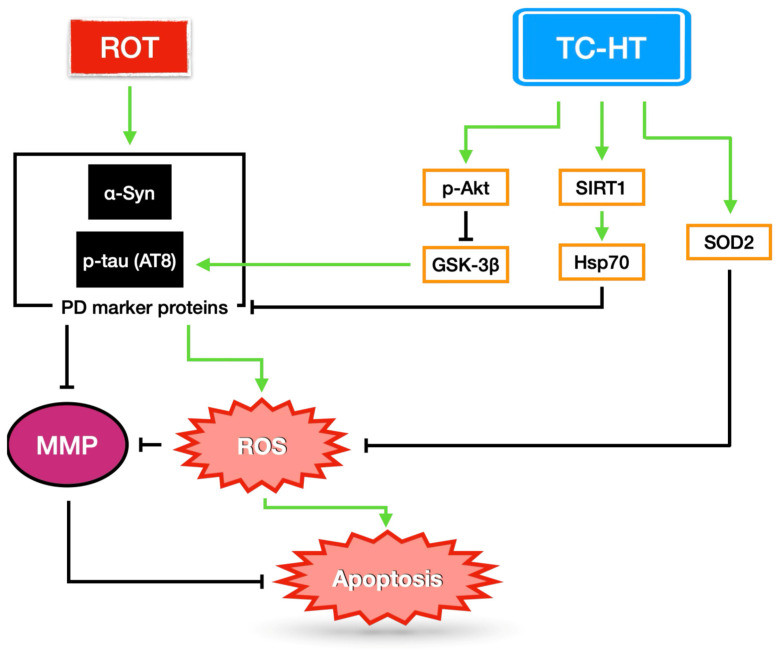
The proposed mechanisms underlying the neuroprotective effect of TC-HT against ROT-induced neural injury. TC-HT suppresses mitochondrial apoptosis and intracellular ROS accumulation by upregulating antioxidative enzyme SOD2 and reducing the expression of PD marker proteins α-syn and p-tau (AT8) through modulating SIRT1/Hsp70 and Akt/GSK-3β signaling pathways.

**Figure 8 ijms-26-06671-f008:**
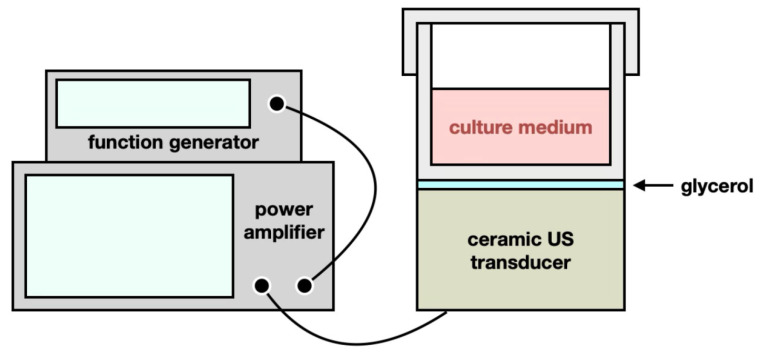
Experimental setups for TC-HT and HT treatments. The US exposure system was composed of a function generator, a power amplifier, and a 0.5 MHz ceramic planar US transducer. The culture plate or dish was placed on the ceramic transducer, with glycerol utilized as the acoustic coupling medium to fill the gap. The transducer delivered acoustic and thermal energy to the SH-SY5Y cells for the administration of TC-HT and HT treatments.

**Figure 9 ijms-26-06671-f009:**
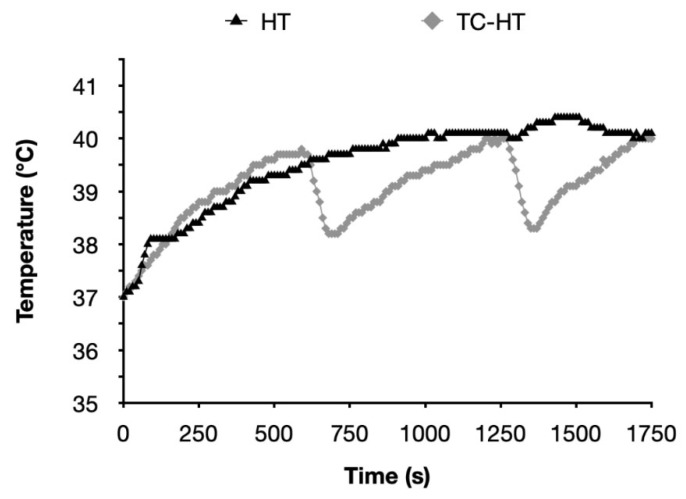
Actual temperatures recorded throughout the TC-HT and HT exposure periods. The actual temperatures in the SH-SY5Y cells were measured using a K-type needle thermocouple at 10 s intervals throughout the exposure durations of TC-HT and HT treatment.

## Data Availability

Data are contained within the article and Appendix A.

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
