# Peer review of "Thermal Cycling-Hyperthermia Attenuates Rotenone-Induced Cell Injury in SH-SY5Y Cells Through Heat-Activated Mechanisms"

_ijms, 2025, doi:10.3390/ijms26146671_

Round 1

Reviewer 1 Report (Previous Reviewer 1)

Comments and Suggestions for Authors

The authors have added relevant content in the discussion section. However, based on the existing literature, aside from the work published by the authors’ own team (References 26–27), there are currently no similar studies on neuroprotection reported by other research groups. Therefore, it would be beneficial for the authors to share more details regarding methodological highlights or practical insights, so that readers may better learn from and apply these techniques. Further elaboration is recommended.

In addition, Figures 1–3 do not provide morphological images of the cells from each group. In these experiments, were there any morphological changes observed in SH-SY5Y cells among the different groups? Specifically, in the manuscript, after 4 hours of TC-HT treatment followed by rotenone (ROT) exposure, what was the morphology of SH-SY5Y cells? Readers may be interested in seeing these morphological outcomes.

Comments on the Quality of English Language

The English could be improved to more clearly express the research.

Author Response

Dear Reviewer 1,

We sincerely appreciate your careful review of our manuscript and your insightful comments. Your suggestions help us improve the clarity of our manuscript and will benefit readers who are unfamiliar with this subject.

The following is a point-by-point response to your comments.

Comment 1:

The authors have added relevant content in the discussion section. However, based on the existing literature, aside from the work published by the authors’ own team (References 26–27), there are currently no similar studies on neuroprotection reported by other research groups. Therefore, it would be beneficial for the authors to share more details regarding methodological highlights or practical insights, so that readers may better learn from and apply these techniques. Further elaboration is recommended.

Response 1:

We thank the reviewer for this careful suggestion. In response to your comments, we have added a new paragraph in the Discussion section that discusses the possible negative effects of continuous heat exposure, and provides additional methodological details on TC-HT, particularly highlighting the importance of the low-temperature interval for optimizing TC-HT effectiveness. We believe this revision will help readers who are new to TC-HT in understanding and utilizing this technique, and the revisions can be found on page 13, lines 370-371, lines 376-385 of the revised manuscript.

Comment 2:

In addition, Figures 1–3 do not provide morphological images of the cells from each group. In these experiments, were there any morphological changes observed in SH-SY5Y cells among the different groups? Specifically, in the manuscript, after 4 hours of TC-HT treatment followed by rotenone (ROT) exposure, what was the morphology of SH-SY5Y cells? Readers may be interested in seeing these morphological outcomes.

Response 2:

We appreciate this valuable suggestion. In response, we have included light microscopy images of SH-SY5Y cells to illustrate their morphological changes following various treatments (Figure 1D, page 4, line 146). The results indicate that ROT exposure significantly disrupted cellular morphology, while pre-treatment with TC-HT 4 hours before ROT exposure preserved the structural integrity. In contrast, continuous HT did not have the same effect. This data more clearly demonstrates the neuroprotective effect of TC-HT against ROT-induced morphological damage in SH-SY5Y cells. The relevant text revisions can be found on lines 141-145 (page 4), lines 151-152 (page 5), and lines 508-511 (page 17) of the revised manuscript.

Finally, we would like to thank you again for your valuable feedback. We believe these revisions have substantially improved the clarity and overall quality of the manuscript.

Reviewer 2 Report (New Reviewer)

Comments and Suggestions for Authors

1. The main message of the manuscript is that with equivalent heating, the pulsed mode is more efficient than the continuous one. The problem is that the modes are not equivalent. In fact, the entire manuscript is based on the fact that at a certain temperature there is parity between the heating and cooling processes. In fact, it is always possible to select a continuous mode that will correspond to the pulsed one in terms of energy and heat engineering. In general, the authors' assertion that the pulsed mode is better than the continuous one is untenable. The authors need to significantly change the rhetoric of the manuscript.

2. In general, heating with ultrasound is uneven in volume. There are always hot and cold areas, this is due to the focusing and unevenness of the ultrasound field. The pulsed mode allows you to prevent the appearance of cold and hot areas, making the temperature higher than average and removing temperature gradients. How do the authors assess the unevenness of heating of the samples? Can the authors provide photographs taken with a thermal imager? For what share of the effect is this phenomenon responsible, in the authors' opinion? 

3. Fig. 9 shows that with constant exposure for almost 30 minutes there is no overheating above 40C, although at certain intervals the temperature rises above 40C. There is no feedback between the energy input and heating. In this regard, overheating may occur sooner or later, since the heating time used in the experiments is 2-3 times longer than in Fig. 9. How can the authors prove that in the system under consideration there is no local and general overheating at exposure times of 60-90 minutes? 

4. It is interesting that at powers of 0.5 W, 1 W and 1.5 W. A classic dependence with one maximum is observed. I will assume that at a power of 0.5 W the heating is insufficient, and at 1.5 W overheating is observed. If we consider the results obtained in the manuscript from this side, then everything is quite explainable, but in this case the study becomes routine...

5. I would like to draw attention to Figure 1. When exposed to ROT, the standard deviations are extremely small. When exposed to ultrasound, the standard deviations are in some cases one order of magnitude larger. How can this be explained? I explain this by the non-uniform ultrasound field and non-uniform heating, which means that many of the postulates of the work need to be revised. Perhaps the authors have an alternative explanation, but I did not find it in the text of the manuscript.

6. The use of ultrasound technologies for local heating is possible, but their widespread use is unlikely. Perhaps this should be written in the text of the manuscript.

Overall, the work is presented with an interesting and unquestionable biological part. At the same time, the formulation of the problem and the conclusions are questionable. Metrology and dosimetry have not been carried out, so only conclusions at the level of laboratory reports can be made (used such and such a mode, got such and such a result). I believe that the manuscript needs to be seriously improved to meet the IJMS standard.

Author Response

Dear Reviewer 2,

We sincerely appreciate your careful review of our manuscript. Your comments have provided us with significant guidance for improving the quality and clarity of our research. We also appreciate your recognition of this study’s biological results. In accordance, we have revised the manuscript to improve its scientific rigor, clarity, and translational significance.

The following is a point-by-point response to your comments.

Comment 1:

The main message of the manuscript is that with equivalent heating, the pulsed mode is more efficient than the continuous one. The problem is that the modes are not equivalent. In fact, the entire manuscript is based on the fact that at a certain temperature there is parity between the heating and cooling processes. In fact, it is always possible to select a continuous mode that will correspond to the pulsed one in terms of energy and heat engineering. In general, the authors' assertion that the pulsed mode is better than the continuous one is untenable. The authors need to significantly change the rhetoric of the manuscript.

Response 1:

Thank you for the insightful comment to the study. We acknowledge that the pulsed mode and the continuous mode are not strictly equivalent in terms of energy delivery and heating profiles, particularly given the millisecond-scale intervals of ultrasound interruption in the pulsed mode. In our experimental design, TC‑HT also uses a continuous ultrasound mode during the high‑temperature stage, but systematically introduces pauses (no ultrasound energy transfer) during the low‑temperature stage. An adequate low-temperature duration (minute-scale) ensures that this intermittent heating approach TC-HT can avoid excessive ROS accumulation to detrimental levels, while still sustaining the signaling transduction of the neuroprotective pathways. This point has been further discussed on Page 13, lines 376-385 of the manuscript. Therefore, under our testing conditions, the intermittent heating method TC-HT exhibited better neuroprotective effect compared the continuously applied heating method HT. We have also revised the manuscript to further clarify the characteristic of the heating methods on:

Page 1, lines 20-21

Page 2, lines 83-85

Page 3, lines 109-111

Page 11, lines 261-262

Page 13, line 376

Page 14, line 423

Page 16, lines 471, 479.

In addition, we also recognize that, in principle, a continuous mode can be adjusted to correspond to a certain pulsed mode, but such precise tuning would require complex simulation and program control, which goes beyond the scope of the present study and is an interesting direction for future investigation.

Comment 2:

In general, heating with ultrasound is uneven in volume. There are always hot and cold areas, this is due to the focusing and unevenness of the ultrasound field. The pulsed mode allows you to prevent the appearance of cold and hot areas, making the temperature higher than average and removing temperature gradients. How do the authors assess the unevenness of heating of the samples? Can the authors provide photographs taken with a thermal imager? For what share of the effect is this phenomenon responsible, in the authors' opinion?

Response 2:

Thank you for this important comment. We agree that non-uniform heating can be an issue when using focused ultrasound; however, in our study, the ultrasound transducer was designed to produce a planar acoustic field. Our aim was to create a controlled in vitro system for reproducible demonstration of TC-HT neuroprotective effects. Additionally, we measured temperatures at multiple positions within the central region (2 cm diameter) of the planar ultrasound transducer (3 cm diameter), which the majority of cells were located. The results showed small temperature fluctuations across this area, as presented in the following table.

While we acknowledge that local temperature fluctuation may still exist and potentially influence cellular responses, our other data, such as mitochondrial function and neuroprotective protein expression, consistently support the overall neuroprotective effects of TC-HT to the treated cells against ROT toxicity. This suggests that once the threshold is surpassed, small thermal fluctuation will not disrupt the activation of neuroprotective signaling pathways and their protective effects. We also included a clarification in the revised manuscript to highlight this consideration in lines 482-483 (Page 16).

Comment 3:

Fig. 9 shows that with constant exposure for almost 30 minutes there is no overheating above 40C, although at certain intervals the temperature rises above 40C. There is no feedback between the energy input and heating. In this regard, overheating may occur sooner or later, since the heating time used in the experiments is 2-3 times longer than in Fig. 9. How can the authors prove that in the system under consideration there is no local and general overheating at exposure times of 60-90 minutes?

Response 3:

Thank you for this detailed observation. We understand the concern regarding potential overheating during prolonged exposure times beyond those shown in Fig. 9. To address this, we continuously monitored the temperature throughout the heating process, where the temperature remained within 40.0 ± 0.5 °C without observed overheating, which was documented in Line 116 (Page 3) and lines 478 (Page 16) of the manuscript. We also revised the manuscript to emphasize this in Line 116 (Page 3) and lines 478 (Page 16). In addition, as shown in Supplementary Figure 1, we have also evaluated the effects of TC‑HT treatment alone (without ROT) on cell viability, which showed no significant difference compared to the untreated control cells. Besides, other data in the manuscript also demonstrated the neuroprotective effect of the TC-HT treatment against ROT. Collectively, we believe that the TC-HT conditions used in our experiments do not induce overheating or detrimental effects to the cells.

Comment 4:

It is interesting that at powers of 0.5 W, 1 W and 1.5 W. A classic dependence with one maximum is observed. I will assume that at a power of 0.5 W the heating is insufficient, and at 1.5 W overheating is observed. If we consider the results obtained in the manuscript from this side, then everything is quite explainable, but in this case the study becomes routine...

Response 4:

Thank you for this thoughtful comment. We agree with your interpretation that the heating at 0.5 W is likely insufficient to activate neuroprotective signaling, while 1.5 W may induce excessive stress or damage. Our results indeed demonstrate a classic bell‑shaped dependence, which suggests an optimal thermal input for achieving neuroprotective effects. Rather than focusing solely on the physical heating parameters, our study uses these findings as a basis to explore how different heating method, intermittent TC-HT and continuous HT, modulate cellular signaling pathways—specifically those related to mitochondrial integrity, ROS generation, and neuroprotective protein signaling. We believe this approach goes beyond routine optimization by revealing the biological response underlying these observations.

Comment 5:

I would like to draw attention to Figure 1. When exposed to ROT, the standard deviations are extremely small. When exposed to ultrasound, the standard deviations are in some cases one order of magnitude larger. How can this be explained? I explain this by the non-uniform ultrasound field and non-uniform heating, which means that many of the postulates of the work need to be revised. Perhaps the authors have an alternative explanation, but I did not find it in the text of the manuscript.

Response 5:

Thank you for this careful observation. We acknowledge that the ultrasound field may exhibit some unevenness, which can contribute to variations in the cellular response. Additionally, biological responses are inherently complex, involving multiple signaling pathways that can differ even under the same stimulation conditions. For example, the represented data in Supplementary Figure 1 showed that cells treated with ultrasound alone can also exhibit small standard deviations in the cell viability results. Moreover, after TC-HT or HT treatment, the overall cellular states, including metabolic activity and intracellular stress, may be altered, which could influence the subsequent response to ROT and potentially result in larger variations. While we agree that the observed variations might be partially related to non-uniform ultrasound fields, we believe that the overall trends and statistically significant differences reported between the experimental groups remain robust.

Comment 6:

The use of ultrasound technologies for local heating is possible, but their widespread use is unlikely. Perhaps this should be written in the text of the manuscript.

Response 6:

Thank you for this thoughtful comment. We agree that while ultrasound technologies can be effective for local heating in vitro or in specific therapies, their broader application, particularly in clinical settings, may face practical limitations. We have already included the difficulties in Lines 386-392 (Page 14) of the Manuscript, and we have also revised the manuscript to remove overly optimistic statements regarding the general applicability of ultrasound-based heating, and emphasize that further research is needed to explore its potential and feasibility in broader applications. The revised parts can be found in lines 403-405 and lines 426-427 (Page 14) of the manuscript.

Finally, we would like to thank you again for your constructive feedback. We believe these revisions have substantially improved the scientific rigor and overall quality of the manuscript.

Reviewer 3 Report (New Reviewer)

Comments and Suggestions for Authors

The manuscript “Thermal Cycling-Hyperthermia Attenuates Rotenone-Induced Cell Injury in SH-SY5Y Cells through Heat-Activated Mechanisms” by Kuo is a research article which examined the neuroprotective effects of thermal cycling-hyperthermia (TC-HT) in an in vitro PD model using rotenone (ROT)-induced human neural SH-SY5Y cells. The authors showed that TC-HT reduced ROT-induced mitochondrial apoptosis and ROS accumulation in SH-SY5Y cells. TC-HT also inhibited the expression of α-syn and p-tau through heat-activated pathways associated with sirtuin 1 (SIRT1) and heat-shock protein 70 (Hsp70) that are involved in protein chaperoning, and caused the phosphorylation of Akt and glycogen synthase kinase-3β (GSK-3β), which inhibit p-tau formation. Thus, the authors indicate that these findings underscore the potential of TC-HT as an effective treatment for PD. In general, this article is critical in this field and contains essential contents. I have minor concerns before this manuscript is accepted for publication.

In bar graphs, all the data plots should be added because the readers can obtain useful information from the data.

For statistical analysis, one-way ANOVA was used. Please add F values in the text or figure legends.

Please add the explanation about how the authors defined the concentration of rotenone!

Author Response

Dear Reviewer 3,

We sincerely appreciate your careful review of our manuscript. Your comments have provided us with significant guidance for improving the data presentation and clarity of our research. We also appreciate your recognition of this study’s results. In accordance, we have revised the manuscript to improve its science rigor and clarity.

The following is a point-by-point response to your comments.

Comment 1:

In bar graphs, all the data plots should be added because the readers can obtain useful information from the data.

Thank you for your suggestion. In response, we have added individual data points as dots to all bar graphs in the revised figures to provide better transparency and allow readers to interpret data distribution more clearly. The revised figures can be found on:

Page 4, Figure 1

Page 6, Figure 2

Page 8, Figure 3

Page 9, Figure 4

Page 10, Figure 5

Page 13, Figure 6

Supplementary Figure 1

Comment 2:

For statistical analysis, one-way ANOVA was used. Please add F values in the text or figure legends.

Response 2:

Thank you for the suggestion. We have added the F-value of each on-way ANOVA in the corresponding figure legends, which can be found on:

Page 4, line 149

Page 5, line 151

Page 6, lines 175, 177-178

Page 8, lines 202-203

Page 9, lines 220-221

Page 10, lines 245-247

Page 13, line 365

Supplementary Figure 1

Comment 3:

Please add the explanation about how the authors defined the concentration of rotenone!

Response 3:

Thank you for your comment. The concentration of rotenone (5 μM) used in this study was selected based on the result in Figure 1A, showing that this concentration induced moderate cytotoxicity in SH-SY5Y cells after 24 h of exposure, and thereby providing an appropriate range to evaluate the protective effects of TC-HT and HT treatment. This concentration has also been widely used in previous studies involving rotenone-induced neurotoxicity in SH-SY5Y cells (as shown below), supporting its relevance as a standard in vitro PD model. We have added a brief explanation and corresponding references to the revised manuscript, which can be found on Page 3, lines 109-111 and Page 11, lines 269-271.

Finally, we would like to thank you again for your constructive feedback. We believe these revisions have substantially improved the scientific rigor and overall quality of the manuscript.

Round 2

Reviewer 1 Report (Previous Reviewer 1)

Comments and Suggestions for Authors

The authors have completed the revision, but issues remain. This study is currently the only one that combines thermal cycling-hyperthermia (TC-HT) with a Parkinson’s disease cell model and validates its neuroprotective effect through the complete SIRT1–Hsp70–Akt–GSK3β signaling pathway. Therefore, as I previously suggested, the methodological details should be carefully described to enable other researchers to replicate the approach in future studies.

Comments on the Quality of English Language

The English could be improved to more clearly express the research.

Author Response

Dear Reviewer 1,

Below, we provide a detailed, point-by-point response to your comments.

Comment 1:

The authors have completed the revision, but issues remain. This study is currently the only one that combines thermal cycling-hyperthermia (TC-HT) with a Parkinson’s disease cell model and validates its neuroprotective effect through the complete SIRT1–Hsp70–Akt–GSK3β signaling pathway. Therefore, as I previously suggested, the methodological details should be carefully described to enable other researchers to replicate the approach in future studies.

Response 1:

We express our sincere gratitude for your feedback and for acknowledging the innovative aspects of our study, particularly the application of TC-HT to a PD cell model and the exploration of the SIRT1–Hsp70–Akt–GSK3β signaling pathway. We concur that methodological transparency is essential for ensuring reproducibility. In the initially submitted manuscript, we have included comprehensive descriptions of our experimental procedures, which encompass:

  • Ultrasound settings, including manufacturer details and US parameters (Pages 15-16, Section 4.2)
  • Temperature monitoring, along with detailed protocols and key parameters for both TC-HT and HT treatments (Pages 16-17, Section 4.3)
  • Cell culture maintenance and conditions for rotenone treatment (Page 15, Section 4.1 and Page 17, Section 4.4)
  • Time points, detailed protocols, dosages of agents, and information regarding the instruments used for each analysis (Pages 17-18, Sections 4.5-4.11)

We believe that these descriptions provide sufficient details for other researchers to replicate our study.

In addition, in response to your previous suggestions, we have incorporated additional texts in the Discussion section that compares our other published TC-HT studies and emphasizes the significant roles of both the intermittent low-temperature intervals (Page 13, Lines 380-388 & 390-391) and the maximum temperature achieved during the high-temperature phase (Pages 14-15, Lines 434-440) in influencing the effects of TC-HT. In this revision, we have also included a discussion (Page 11, Lines 274-281) that highlights the importance of selecting an appropriate heating temperature for the specific experimental model, as well as a clearer explanation of why extending the low-temperature interval beyond its optimal duration may negate the beneficial effects of TC-HT (Page 14, Lines 386-390). These revisions underscore the significance of key parameters in TC-HT and offer guidance for further optimization of TC-HT to enhance its effectiveness in the in vitro ROT-induced PD model, as well as its application to other in vitro or in vivo disease models.

However, we genuinely value your comments and would appreciate further clarification regarding which specific aspects of the methodology you find lacking, so that we may address your concerns more directly and ensure that the manuscript aligns with your expectations.

Finally, we would like to reiterate our gratitude for your insightful comments and would welcome the opportunity to address them further.

Reviewer 2 Report (New Reviewer)

Comments and Suggestions for Authors

Interesting biophysical research, but the level of metrology and presentation of physical processes is clearly insufficient. Minor revision, try to make improvements to the text of the manuscript.

Author Response

Dear Reviewer 2,

Thank you for recognizing the biophysical aspects of our study and for providing constructive comment to help us improve the manuscript. In accordance with your suggestions, we have revised the manuscript to better meet your expectations.

The following is the point-by-point response to your comments.

Comment 1:

Interesting biophysical research, but the level of metrology and presentation of physical processes is clearly insufficient. Minor revision, try to make improvements to the text of the manuscript.

Response 1:

We appreciate this thoughtful comment. In response, we have revised the manuscript to more explicitly indicate that the maximum temperature attained during the TC-HT or HT conditions was 40 ± 0.5 °C, which was achieved through US application at an intensity of 1.0 W. These revisions can be found in:

Page 11, Line 278

Page 13, Line 376

Page 15, Line 440

Page 16, Line 488

Additionally, in accordance with your previous suggestions, we have revised the text to present the laboratory findings in a more factual manner, specifically by detailing the applied methodology and directly reporting the associated outcomes without overinterpretation). These revisions have been included in:

Page 3, Line 103

Page 14, Lines 415, 417, 418, 429-430

To enhance the illustration of the physical characteristics of our US exposure system, we have added a description of the temperature fluctuation assessment (Page 16, Lines 490-494) and included the relevant results in Supplementary Figure 2.

Finally, we would like to express our sincere appreciation once again for your invaluable comments. We believe that these revisions have significantly improved the clarity, transparency, and reproducibility of the manuscript.

Round 3

Reviewer 1 Report (Previous Reviewer 1)

Comments and Suggestions for Authors

The revised manuscript has been completed, but the following issues remain:

  1. In the third revised version of the manuscript, the authors did not modify or provide further details regarding the methodology. The overall experimental procedure remains similar to that described in the team's previously published reference 26. My concern is whether, based on the current description, readers would be able to reliably reproduce the experiment.
  2. The experimental design or methodology is similar to that of previous publications by the same team (e.g., references 26 and 47). However, it lacks a control group to evaluate the effects of thermal cycling hyperthermia (TC-HT) alone on SH-SY5Y cells. In other words, it remains unclear whether TC-HT induces non-specific cellular damage in SH-SY5Y cells.
  3. There is a lack of positive control or comparative treatment groups. It is recommended to include agents such as antioxidants (e.g., NAC), SIRT1 activators, or Hsp70 inducers as positive controls.
Comments on the Quality of English Language

The English could be improved to more clearly express the research.

Author Response

Dear Reviewer 1,

Thank you for the comments and feedback. Below, we provide a detailed, point-by-point response to your comments.

Comment 1:

In the third revised version of the manuscript, the authors did not modify or provide further details regarding the methodology. The overall experimental procedure remains similar to that described in the team's previously published reference 26. My concern is whether, based on the current description, readers would be able to reliably reproduce the experiment.

Response 1:

Thank you for this concern. While we acknowledge your point, we respectfully disagree. The TC-HT procedure presented in this study shares the same conceptual framework as in our previous work (Ref. 26): a repetitive high- and low-temperature exposure procedure, which can be easily applied and optimized.

In addition, the study differs from Ref. 26 in both the disease model and the heat exposure system. As described and discussed in the manuscript, different models require re-optimization of parameters. In the revised manuscript, we have clearly provided detailed information regarding the manufacturer and specifications of the function generator, power amplifier, and ultrasound transducer used in this study, as well as the US signal parameters, final US output intensity, and the actual cell temperature achieved. Importantly, we emphasized in the manuscript that achieving the similar cell temperature is more critical than replicating the exact device or hardware setup, allowing researchers flexibility in replication. We believe the descriptions and guidance are enough to reproduce the TC-HT and HT treatment in this study.

Furthermore, across the submission history, none of the other 4 reviewers raised similar concerns regarding TC-HT methodological clarity after the latest revisions. This suggests that the methodology descriptions have been sufficiently improved to meet peer-review standards. The TC-HT treatment has also been successfully applied to other models, including AD in vitro (Ref. 26), in vivo AD model, cancer cell studies, and immunomodulation experiments, demonstrating its widespread use and recognition in the scientific community. Based on these points, we believe no further revision is required for the TC-HT methodology.

Comment 2:

The experimental design or methodology is similar to that of previous publications by the same team (e.g., references 26 and 47). However, it lacks a control group to evaluate the effects of thermal cycling hyperthermia (TC-HT) alone on SH-SY5Y cells. In other words, it remains unclear whether TC-HT induces non-specific cellular damage in SH-SY5Y cells.

Response 2:

Thank you for raising this issue. The data you refer to had already provided in the supplementary files, which were attached in our response to the other reviewer in the first-round revision of the previous submission (manuscript ID: ijms-3556577, submitted at 2025/03/14). The data clearly showed that TC-HT treatment alone did not induce non-specific cytotoxicity in SH-SY5Y cells, and the results were appropriately described and integrated into the revised manuscript text during that revision.

Comment 3:

There is a lack of positive control or comparative treatment groups. It is recommended to include agents such as antioxidants (e.g., NAC), SIRT1 activators, or Hsp70 inducers as positive controls.

Response 3:

We appreciate your suggestion. However, the main objective of this study is to investigate the neuroprotective effects of TC-HT against ROT-induced cytotoxicity in an in vitro PD model and to explore the heat-associated mechanisms activated by TC-HT treatment. While incorporating positive controls such as antioxidants or pathway-specific activators may provide additional mechanistic insights, it would shift the study beyond its current scope. Your recommendation is valuable, and will be considered in the future studies aiming to investigate specific molecular contributions of each pathway.

Final Remarks:

Across the submission history, we valued your comments and have worked diligently to revise the manuscript accordingly. We have clarified the TC-HT methodology including the instrument specifications and physical parameters, distinguished this study from Ref. 26, and added guidance discussion to support future replication and optimization. However, despite these efforts, concerns regarding methodology were repeatedly raised without clear specification. Meanwhile, none of the other 4 reviewers have expressed further concerns regarding the methodology after the latest revisions.

Additionally, and to our surprise, Comment 2 requests data that were already provided in a revision submitted in the previous submission (ijms-3556577). Moreover, the entire review report in Round 1 of this submission is identical to the one in Round 4 of the previous submission (ijms-3556577), and that a cover letter to respond the comments in Round 4 were attached with the resubmitted manuscript of the submission. These strongly raise a concern that our manuscript may not have been carefully reviewed.

Given the significant effort we have made to revise and clarify the manuscript, we are concerned that the feedback received in this round does not reflect a fair or thorough evaluation. We believe that the current version of the manuscript has met the standards for publication in IJMS and is in its final form, ready for acceptance.

This manuscript is a resubmission of an earlier submission. The following is a list of the peer review reports and author responses from that submission.

Round 1

Reviewer 1 Report

Comments and Suggestions for Authors

This manuscript has the following issues:

Abstract: While the study's relevance to PD treatment is mentioned, it could more explicitly link these findings to potential clinical applications or future research directions (involving in vivo models or human trials).

Introduction: While briefly mentioning the potential risks of heat therapy, it lacks a thorough discussion on how these risks could impact the practical application of the treatment or how they might be mitigated or managed in future research.

Results:

  1. It is suggested to use gray scale for histograms in Figures 1-6.

  2. A summary figure should be included to help readers understand the interconnections of the results more easily.

Discussion:

  1. The discussion of the limitations of this study appears insufficient.

  2. There is a lack of feasibility analysis for clinical applications. Although the potential clinical applications of TC-HT are mentioned, there is a lack of discussion on the challenges and obstacles that might be encountered in a clinical setting when implementing this treatment strategy.

  3. There is a lack of statement on future research directions, such as exploring different temperature settings, treatment durations, or combinations with other treatment methods.

Methods:

  1. In the western blot analysis, the dilution ratios of primary and secondary antibodies should be provided.

  2. Why do the temperature and duration of each cycle in the TC-HT method and shown in Figure 8 differ from those in Reference 26? What is the reason for this discrepancy? How should other researchers set these parameters when conducting this research?

Comments on the Quality of English Language

The English could be improved to more clearly express the research.

Author Response

Dear Reviewer 1,

We express our sincere gratitude for your comprehensive and constructive review of our manuscript. Your insightful comments have offered us significant perspectives and direction, and we appreciate your recognition of the importance of our research in the context of Parkinson’s disease (PD). In response to your comments and suggestions, we have made substantial revisions to the manuscript to enhance its clarity, scientific rigor, and potential translational applicability.

The following is a point-by-point response to your comments.

Comment 1:

Abstract: While the study's relevance to PD treatment is mentioned, it could more explicitly link these findings to potential clinical applications or future research directions (involving in vivo models or human trials).

Response 1:

We appreciate this constructive suggestion. Accordingly, we have refined the concluding sentence of the Abstract to more clearly elucidate the possible use of FUS-based TC-HT in in vivo studies and potential human trials.

This change can be found on page 1, lines 27–29.

Comment 2:

Introduction: While briefly mentioning the potential risks of heat therapy, it lacks a thorough discussion on how these risks could impact the practical application of the treatment or how they might be mitigated or managed in future research.

Response 2:

Thank you for raising this important point. We acknowledge that the potential risks of continuous heat exposure deserve more discussion, particularly regarding their application for clinical translation.

Therefore, we have revised the relevant sentences to more clearly state how risks may impact clinical trials, and how they could potentially be alleviated through the use of programmable FUS in the future.

This revision sentence can be found on page 2, lines 76–79 and lines 86-90 of the revised manuscript.

Comments 3:

Results:

  1. It is suggested to use gray scale for histograms in Figures 1-6.

Response 3:

Thank you for your suggestion. We agree with this comment and understand that using gray scale for bar plots can improve figure clarity.

Thus, we have modified all histograms in Figures 1-6 to use gray scale for bar representations. The updated figures can be found on page 4 (Figure 1), page 5 (Figure 2), page 7 (Figure 3), page 8 (Figure 4), page 9 (Figure 5), and page 12 (Figure 6), of the revised manuscript. The revised figure files have been uploaded accordingly.

Comments 4:

  1. A summary figure should be included to help readers understand the interconnections of the results more easily.

Response 4:

We are grateful for this suggestion and found that including a summary figure can effectively illustrate the interconnections among our key findings and enhance the clarity of the conclusion.

We have added a new summary figure (now Figure 7), which presents the proposed mechanisms underlying the neuroprotective effects of TC-HT against ROT-induced neural injury. The figure caption has also been added on page 13, and referenced on page 13, Line 387, accordingly.

Comment 5:

Discussion:

  1. The discussion of the limitations of this study appears insufficient.

Response 5:

Thank you for pointing this out. It is important to recognize that the present study is limited to an in vitro model and functions primarily as a preliminary exploration of the potential neuroprotective properties of TC-HT, rather than providing definitive evidence for clinical application.

Therefore, we have revised Discussion to explicitly state that further validation through in vivo PD models is essential to determine whether TC-HT can genuinely enhance disease-related behavioral symptoms and mitigate the accumulation of pathological markers in animal brains.

The revision sentences can be found on page 12, lines 352–357 of the revised manuscript.

Comment 6:

  1. There is a lack of feasibility analysis for clinical applications. Although the potential clinical applications of TC-HT are mentioned, there is a lack of discussion on the challenges and obstacles that might be encountered in a clinical setting when implementing this treatment strategy.

Response 6:

Thank you for this valuable comment. We agree that evaluating the feasibility of TC-HT for clinical applications is essential. In response, we have revised the Discussion section to include a more detailed analysis of the potential challenges associated with the translation of TC-HT into clinical practice, especially in the context of its application through FUS systems. We have also further discussed how FUS-based TC-HT may address these challenges by utilizing magnetic resonance imaging-guided FUS systems that incorporate thermometry, thereby facilitating localized energy delivery and mitigating the risk of overheating.

These revisions can be found on page 12, lines 362–369 of the revised manuscript.

Comment 7:

  1. There is a lack of statement on future research directions, such as exploring different temperature settings, treatment durations, or combinations with other treatment methods.

Response 7:

Thank you for this suggestion. We agree that outlining future research directions would enhance the completeness and translational relevance of our study.

Accordingly, we have revised Discussion section to include a new statement describing potential future investigations. These include the examination of various temperature settings and treatment durations to optimize the efficacy of TC-HT, as well as the assessment of the synergistic effects of TC-HT in conjunction with PD drugs or alternative therapeutic strategies.

The revised content can be found on page 12-13, lines 370–373 of the revised manuscript.

Comment 8:

Methods:

  1. In the western blot analysis, the dilution ratios of primary and secondary antibodies should be provided.

Response 8:

Thank you for pointing this out. We agree with this comment and understand the importance of clearly specifying antibody dilution ratios to ensure reproducibility.

We have added the dilution ratios of all primary and secondary antibodies used in the western blot analysis. These revisions can be found in page 16, lines 507-514.

Comment 9:

  1. Why do the temperature and duration of each cycle in the TC-HT method and shown in Figure 8 differ from those in Reference 26? What is the reason for this discrepancy? How should other researchers set these parameters when conducting this research?

Response 9:

Thank you for raising this important question. The difference in temperature and cycle durations between our current study and the previous research (Reference 26) can be attribuated to the utilization of different heat delivery methods. In the prior study, TC-HT was administered through a conductive heating system, whereas in the present study, TC-HT was delivered via an ultrasound-based heating system designed to replicate FUS stimulation. Additionally, the neurotoxic agents used were different; the previous study employed hydrogen peroxide and Aβ, while the current study utilized rotenone. These differences required re-optimization of TC-HT parameters to ensure effective neuroprotection in the rotenone-induced in vitro PD model.

To support reproducibility and facilitate future research, we have provided a comprehensive temperature profile of the optimized TC-HT settings in Figure 9 (original Figure 8) in the manuscript, which illustrates the real-time temperature curve as sensed by the cells in our system. We suggest that future studies aiming to replicate this methodology should calibrate their systems to achieve comparable cell temperatures, rather than directly adopting the ultrasound settings.

Finally, we would like to thank you again for your valuable comments. We believe that these revisions have significantly enhanced the manuscript and improved its overall quality.

Reviewer 2 Report

Comments and Suggestions for Authors

Dear authors,

I have read and reviewed the study “Thermal Cycling-Hyperthermia Attenuates Rotenone-Induced 2 Cell Injury in SH-SY5Y Cells through Heat-Activated Mechanisms”. Overall, it is very interesting and innovative study. Here are my specific suggestions:

  1. Lines 32-33 – please add the years of the data
  2. Line 35 „Besides aging, exposure to pesticides such as rotenone (ROT) can also induce PD-like symptoms” – please add citation and the relevant concentrations
  3. Line 44 – please specify which animal model
  4. Lines 89-98 – I would keep them for conclusions
  5. Also, I would recommend the introductions of a short phrase regarding the SH-SY5Y cells in the introduction section
  6. Please fully revise the results section and simplify it, keep the quotes and additional sentences only for the discussion section
  7. Figure 1 B and C – please state the concentration of ROT exposed
  8. Similar for the next figures
  9. If you did not state the exact p value within the text (which I recommend instead of semi-combining the results with discussion) please add it on the graphics, also explain “N.S.”
  10. Please improve the discussion section with the aforementioned unnecessary sentences from results section and harmonize the text
  11. I noticed that you are very confident about the capacity of HT as a key therapy for PD, but certainly specific studies are needed in this regard before we can support this with such certainty. To avoid potential issues, please mention the limitations of the study and what could be optimized to truly be considered a real therapy for PD from a clinical point of view.
  12. Line 462 please add the manufacturer and country
  13. Line 493 please add the software version used

Author Response

Dear Reviewer 2,

We are grateful for your comprehensive and insightful review of our manuscript. Your comments have provided us with significant guidance for improving the quality and clarity of our research. We also appreciate your recognition of this study’s importance within the context of Parkinson’s disease (PD) research. In accordance with your suggestions, we have meticulously revised the manuscript to improve its scientific rigor, clarity, and translational significance.

The following is a point-by-point response to your comments.

Comment 1:

Lines 32-33 – please add the years of the data

Response 1:

Thank you for your observing the detail. We have added the years of the cited data. This revision can be found on page 1, lines 35 of the revised manuscript.

Comment 2:

Line 35 Besides aging, exposure to pesticides such as rotenone (ROT) can also induce PD-like symptoms” – please add citation and the relevant concentrations

Response 2:

Thank you for pointing out this question. We have made the revisions to include appropriate citations supporting the use of ROT as a neurotoxicant that induces PD-like symptoms in both in vitro and in vivo models. The relevant concentrations used in these references have also been briefly mentioned.

This revision can be found on page 1, line 39 and page 2, line 45 of the revised manuscript.

Comment 3:

Line 44 – please specify which animal model

Response 3:

Thank you for the suggestion. We have specified the type of animal model employed in the referenced study, which can be found on page 2, line 45 of the revised manuscript.

Comment 4:

Lines 89-98 – I would keep them for conclusions

Response 4:

We appreciate your suggestion. In response to your comments, we have reconstructed the paragraph to concentrate exclusively on the experimental design and objectives of this study, eliminating any statements pertaining to results and interpretations. This revision can be found on page 2-3, lines 91-101 of the revised manuscript.

In addition, the key points originally presented in this paragraph are already addressed in the concluding section of the Discussion (Page 12-13, lines 350-366 of the original manuscript), thus no further modifications were deemed necessary for that section.

Comment 5:

Also, I would recommend the introductions of a short phrase regarding the SH-SY5Y cells in the introduction section

Response 5:

Thank you for the suggestion. We have added a concise description of the SH-SY5Y cell line in the Introduction section to elucidate its significance in Parkinson’s disease research. Specifically, we have indicated that SH-SY5Y is a human neuroblastoma cell line that demonstrates dopaminergic properties and is extensively utilized as an in vitro model for investigating PD-related neurotoxicity and protective strategies. This revision can be found on pages 1-2, lines 92–93 of the revised manuscript.

Comment 6:

Please fully revise the results section and simplify it, keep the quotes and additional sentences only for the discussion section

Response 6:

Thank you for this valuable suggestion. In response to your suggestions, we have revised the Results section to focus solely on the descriptive reporting of experimental findings, with interpretative or conclusive statements now incorporated into the Discussion.

The line numbers for the revised content in the Discussion will be detailed in Response 10.

Comment 7:

Figure 1 B and C – please state the concentration of ROT exposed

Response 7:

Thank you for your suggestion. While the concentration of 5 μM ROT used for subsequent experiments was mentioned in the description of Figure 1A, we concur that reiterating this information in the corresponding sentences for Figures 1B and 1C would enhance clarity.

Therefore, we have included this concentration on Pages 3, lines 111–112, line 128, and line 131 of the revised manuscript, indicating that the experiments were conducted under 5 μM ROT exposure. Additionally, we have also updated the figure caption of Figure 1 (Page 4, line 140) to include this concentration.

Comment 8:

Similar for the next figures

Response 8:

Thank you for the follow-up comment. We have reviewed all subsequent figures to ensure that the concentration of ROT (5 μM) is clearly stated in both the main text descriptions and the corresponding figure captions, thereby promoting consistency and clarity across all data presentations.

The revision texts can be founded on:

Page 4, line 148

Page 5, line 151

Page 6, line 166, line 174, line 180

Page 7, line 190, line 196

Page 8, line 202, line 207, line 221, line 227

Page 9, line 233

Page 11, line 338

Page 12, line 346

Comment 9:

If you did not state the exact p value within the text (which I recommend instead of semi-combining the results with discussion) please add it on the graphics, also explain “N.S.”

Response 9:

Thank you for this helpful suggestion. In response to your comments, we have reviewed all statistical comparisons and have included the exact p values for each group comparison in the text to enhance clarity and precision in the presentation of our results.

Moreover, we have updated the figure captions to provide a clear explanation of “N.S.”, which denotes “not significant”.

These revisions can be found in the Results section throughout the manuscript, as well as in the captions of Figures 1–6.

Comment 10:

Please improve the discussion section with the aforementioned unnecessary sentences from results section and harmonize the text.

Response 10:

In continuation of Response 6, we have revised the Discussion section to integrate pertinent interpretative content that was previously situated within the Results section. These enhancements offer a more detailed comprehensive interpretation of our data, thereby elucidating the potential implications of TC-HT. Additionally, the text has been refined for clarity and consistency throughout the revised manuscript.

The revision texts can be founded on:

Page 10, lines 255-256, lines 266-268, lines 271-273

Page 11, lines 292-294

Comment 11:

I noticed that you are very confident about the capacity of HT as a key therapy for PD, but certainly specific studies are needed in this regard before we can support this with such certainty. To avoid potential issues, please mention the limitations of the study and what could be optimized to truly be considered a real therapy for PD from a clinical point of view.

Response 11:

Thank you for this important and insightful comment. We concur that the current findings predominantly underscore the neuroprotective effects of TC-HT in vitro, necessitating further investigations before it can be considered as a viable therapeutic strategy for PD. In response to your suggestion, we have implemented several revisions throughout the manuscript to address the study’s limitations, the feasibility of clinical translation, and prospective directions for future research.

Specifically, we have:

  • Acknowledged the in vitro limitation of the current study, and emphasized the need for in vivo validation to assess behavioral effects and protein accumulation outcomes, which would serve as a foundation for future potential clinical trials. (Page 12, lines 352–357)
  • Discussed issues related to clinical feasibility, including challenges such as the precision of thermal delivery and patient-specific variability, and proposed that the integration of magnetic resonance imaging-guide FUS with thermometry may assist in overcoming these challenges. (Page 12, lines 362–369)
  • Outlined future research directions, which include optimizing temperature and duration settings, as well as investigating the combination of TC-HT with established PD medications or other therapeutic strategies. (Pages 12–13, lines 370–373)

Comment 12:

Line 462 please add the manufacturer and country

Response 12:

We appreciate your attention to the detail. The information regarding the manufacturer and country of origin for the confocal microscopy utilized in the study has been incorporated. This revision can be found in the revised manuscript on Page 16, Lines 491-492.

Comment 13:

Line 493 please add the software version used

Response 13:

Thank you for pointing out this question. We have added the software version used in our analysis, and the information can be found in the revised manuscript on Page 17, Line 524.

Finally, we would like to express our gratitude again for your valuable comments. We believe that these revisions have substantially strengthened the manuscript and improved its overall quality.

Reviewer 3 Report

Comments and Suggestions for Authors

The paper evaluated the neuroprotective effects of cycling hyperthermia (TC-HT) against oxidative stress and neurotoxicity induced by rotenone (ROT) in human SH-SY5Y cells. In particular, TC-HT reduced apoptosis and the accumulation of reactive oxygen species induced by ROT. Further, TC-HT inhibited the expression of α-synuclein and phosphorylated tau through heat-activated pathways associated with sirtuin 1 and heat shock protein 70. This process, which is involved in protein chaperoning, also led to the phosphorylation of Akt and glycogen synthase kinase-3β.

The paper evaluated several cellular pathways to demonstrate the neuroprotective effects of TC-HT. All the cellular pathways probably share the phenomena of neurohormones.

The authors should improve the reported data. Below are some issues to address during the revision.

Major

  1. The authors reported the neurotoxicity of ROT to determine the appropriate concentration for the neuroprotective assay. They should also show the neurotoxicity of various intensities of TC-HT and HT used alone, without treatment, to evaluate the impact of heat on these cells.

  1. The authors demonstrated the antioxidant activity of heat treatment by measuring levels of reactive oxygen species and superoxide dismutase activity (SOD). This antioxidant activity is likely a key factor behind the neuroprotective effects of the treatment. However, they should explore mechanisms that are not solely mediated by SOD. It is important to evaluate additional biological endpoints related to the activation of endogenous antioxidant mechanisms, such as catalase and glutathione, which are ultimately involved in activating the Nrf2/ARE pathway. Notably, mild heat can stimulate this pathway—for example, mild heat shock at 40 °C has been shown to increase autophagy levels (as referenced in Cell Stress Chaperones, August 2024, Volume 29, Issue 4, pages 567-588, doi: 10.1016/j.cstres.2024.06.001). Following this, the discussion focuses on the potential activation of this redox pathway.

Minor

  1. The authors highlight that focused ultrasound can implement TC-HT in the abstract. However, they only discussed this aspect without presenting data to demonstrate the efficacy of the treatment. Therefore, they should remove this hypothesis from the abstract.

Author Response

Dear Reviewer 3,

We sincerely appreciate the reviewer’s comprehensive evaluation of our manuscript. Your insightful feedback has offered us valuable perspectives that enhance the clarity, scientific rigor, and overall quality of our research. We are particularly appreciative of your acknowledgment of the neuroprotective potential associated with thermal cycling hyperthermia (TC-HT) and your suggestions for further mechanistic investigation and safety assessment.

The following is a point-by-point response to your comments.

Comment 1:

Major

The authors reported the neurotoxicity of ROT to determine the appropriate concentration for the neuroprotective assay. They should also show the neurotoxicity of various intensities of TC-HT and HT used alone, without treatment, to evaluate the impact of heat on these cells.

Response 1:

Thank you for this important suggestion. In response to your suggestions, we have performed additional experiments to assess the neurotoxicity associated with various intensities of TC-HT and HT applied independently, without ROT treatment. The finding from these experiments help to clarify the direct impact of thermal stimulation on the viability of SH-SY5Y cells. These results have been incorporated into the revised manuscript and are presented as Supplementary Figure 1, and the numerical data have been included in the updated Minimal Dataset in Supplementary Files. Corresponding descriptions have also been added to Results on page 3, lines 114–120.

Comment 2:

The authors demonstrated the antioxidant activity of heat treatment by measuring levels of reactive oxygen species and superoxide dismutase activity (SOD). This antioxidant activity is likely a key factor behind the neuroprotective effects of the treatment. However, they should explore mechanisms that are not solely mediated by SOD. It is important to evaluate additional biological endpoints related to the activation of endogenous antioxidant mechanisms, such as catalase and glutathione, which are ultimately involved in activating the Nrf2/ARE pathway. Notably, mild heat can stimulate this pathway—for example, mild heat shock at 40 °C has been shown to increase autophagy levels (as referenced in Cell Stress Chaperones, August 2024, Volume 29, Issue 4, pages 567-588, doi: 10.1016/j.cstres.2024.06.001). Following this, the discussion focuses on the potential activation of this redox pathway.

Response 2:

We sincerely appreciate this insightful suggestion. We agree that antioxidant mechanisms beyond SOD2 should be considered to provide a more comprehensive understanding of TC-HT–mediated neuroprotection. Accordingly, we have revised Discussion section to incorporate additional description on the potential involvement of other endogenous antioxidant pathways. In particular, we have emphasized the possible activation of the Nrf2/ARE signaling pathway by heat, which is known to regulate a diverse array of antioxidant genes. Relevant references to support this perspective have also been included. These additions suggest that TC-HT may exert its antioxidative effects through a more extensive redox-regulatory network.

The revised discussion can be found on page 10, lines 283–291 of the revised manuscript.

Comment 3:

Minor

The authors highlight that focused ultrasound can implement TC-HT in the abstract. However, they only discussed this aspect without presenting data to demonstrate the efficacy of the treatment. Therefore, they should remove this hypothesis from the abstract.

Response 3:

Thank you for pointing this out. We acknowledge that the original phrasing in the Abstract may have inadvertently suggested that focused ultrasound (FUS) has already been established as a delivery method for TC-HT, which is beyond the scope of the current in vitro study. In response to this concern, we have revised the sentence to avoid this overstatement while still adhering to Reviewer 1’s suggestion to more explicitly convey the potential clinical relevance and future research directions related to this work.

The revised sentence now reads:

These findings underscore the potential of TC-HT as an effective treatment for PD in vitro, supporting its further investigation in in vivo models or potentially in human trials, with focused ultrasound (FUS) as a feasible heat-delivery approach.

This change clarifies that FUS is proposed as a prospective method for future TC-HT applications, rather than as a validated strategy. The revised sentence appears on page 1, lines 27–29 of the revised manuscript.

Finally, we express our gratitude for your insightful comments and constructive suggestions. We believe that the revisions made in response to your comments have significantly enhanced the clarity, comprehensiveness, and overall quality of the manuscript.

Round 2

Reviewer 1 Report

Comments and Suggestions for Authors

The authors have completed the revisions; however, the following issues remain:

  1. In the abstract, it should be clearly stated that the pre-treatment with thermal cycling-hyperthermia (TC-HT) confers neuroprotective effects in the rotenone (ROT)-induced in vitro Parkinson’s disease model using human SH-SY5Y neuronal cells, in order to avoid potential misunderstandings by the readers.
  2. The authors should incorporate the content of Response 9 into the Discussion section, to facilitate readers’ understanding and support the reproducibility of the TC-HT experiments.
Comments on the Quality of English Language

The English could be improved to more clearly express the research.

Author Response

Dear Reviewer 1,

We sincerely appreciate your careful review of our manuscript and your insightful comments. Your suggestions have significantly contributed to the clarification of essential elements of the study and enhanced the accuracy of our statements in the study.

The following is a point-by-point response to your comments.

Comment 1:

In the abstract, it should be clearly stated that the pre-treatment with thermal cycling-hyperthermia (TC-HT) confers neuroprotective effects in the rotenone (ROT)-induced in vitro Parkinson’s disease model using human SH-SY5Y neuronal cells, in order to avoid potential misunderstandings by the readers.

Response 1:

We appreciate this constructive suggestion. In response to your comments, we have revised the Abstract to specify that the neuroprotective effects of TC-HT pre-treatment were demonstrated in ROT-induced SH-SY5Y cells, which serve as an in vitro model for Parkinson’s disease. This modification aims to mitigate any potential misinterpretations by readers. The revised text can be found on page 1, lines 22–25 of the manuscript.

Comment 2:

The authors should incorporate the content of Response 9 into the Discussion section, to facilitate readers’ understanding and support the reproducibility of the TC-HT experiments.

Response 2:

We appreciate your suggestion and concur that integrating this information into the Discussion section will enhance readers’ comprehension and support the reproducibility of the TC-HT methodology. Thus, we have included additional statements to elucidate the rationale for re-optimizing the TC-HT parameters in this study, emphasizing that achieving comparable cellular temperature profiles is of greater importance than merely matching the ultrasound settings.

These revisions can be found on page 10, lines 257–259 and page 13, lines 395–397 of the revised manuscript.

Finally, we would like to thank you again for your valuable comments. We believe that these revisions have substantially strengthened the manuscript and enhanced its overall quality.

Reviewer 2 Report

Comments and Suggestions for Authors

Congratulations on the work you have done and good luck for the future.

Author Response

Comment:

Congratulations on the work you have done and good luck for the future.

Response:

We sincerely thank the reviewer for the kind words and encouragement. We truly appreciate your positive feedback and support for our work.

Reviewer 3 Report

Comments and Suggestions for Authors

The manuscript is suitable for publication.

Author Response

Comment:

The manuscript is suitable for publication.

Response:

We truly appreciate your kind recommendation and are grateful for your support for the publication of our manuscript.

Round 3

Reviewer 1 Report

Comments and Suggestions for Authors

The response to Question 2 [L395-397: “Future studies aiming to replicate this methodology should calibrate their systems to achieve comparable cell temperatures, rather than directly adopting the ultrasound settings.”] is still not sufficiently scientific or patient. While this may be clear to the authors—since the current study and reference 26 come from the same research team—it may be difficult for readers who are encountering this topic for the first time to understand. I kindly ask the authors to further elaborate and supplement this part of the manuscript.

Comments on the Quality of English Language

The English could be improved to more clearly express the research.

Author Response

Dear Reviewer 1,

We sincerely appreciate your careful review of our manuscript and your insightful comments. Your suggestion helped us improve the clarity of our manuscript and could avoid potential misunderstanding in the study.

The following is a point-by-point response to your comments.

Comment 1:

The response to Question 2 [L395-397: “Future studies aiming to replicate this methodology should calibrate their systems to achieve comparable cell temperatures, rather than directly adopting the ultrasound settings.”] is still not sufficiently scientific or patient. While this may be clear to the authors—since the current study and reference 26 come from the same research team—it may be difficult for readers who are encountering this topic for the first time to understand. I kindly ask the authors to further elaborate and supplement this part of the manuscript.

Response 1:

We appreciate this careful suggestion. In response, we have expanded the statement in the Discussion section to emphasize that the temperature is a more important factor in replicating the neuroprotective effects of TC-HT rather than the ultrasound parameters or the heat delivery method. This revision can improve the clarity for readers who may be less familiar with this topic, and can be found on page 13, lines 395–402 of the manuscript.

Finally, we would like to thank you again for your insightful comments. We believe these revisions have substantially improved the clarity of the manuscript and its overall quality.

Round 4

Reviewer 1 Report

Comments and Suggestions for Authors

The authors have added relevant content in the discussion section. However, based on the existing literature, aside from the work published by the authors’ own team (References 26–27), there are currently no similar studies on neuroprotection reported by other research groups. Therefore, it would be beneficial for the authors to share more details regarding methodological highlights or practical insights, so that readers may better learn from and apply these techniques. Further elaboration is recommended.

In addition, Figures 1–3 do not provide morphological images of the cells from each group. In these experiments, were there any morphological changes observed in SH-SY5Y cells among the different groups? Specifically, in the manuscript, after 4 hours of TC-HT treatment followed by rotenone (ROT) exposure, what was the morphology of SH-SY5Y cells? Readers may be interested in seeing these morphological outcomes.

Comments on the Quality of English Language

The English could be improved to more clearly express the research.